# Reward-Guided Prompt Evolving in Reinforcement Learning for LLMs

**Ziyu Ye**[12] **Rishabh Agarwal**[1] **Tianqi Liu**[1] **Rishabh Joshi**[1] **Sarmishta Velury**[1] **Quoc V. Le**[1] **Qijun Tan**[1] **Yuan Liu**[1]

## Abstract

Existing reinforcement learning (RL) methods for large language models (LLMs) rely on static prompt sets, where prompts are curated *a priori*, and sampled in a fixed schedule for training, regardless of their usefulness to the RL process.

We design `eva`, the first method that allows LLMs to **prioritize** and **adaptively create** useful prompts during RL training by *reward signals*.

In principle, `eva` (Evolving *via* Asymmetric Self-Play) casts language model training as a game between: (1) a **creator**, who samples and generates training prompts, and (2) a **solver**, who generates responses to the prompts.

`eva` is simple, suits both offline and online RL for LLMs, and sets a new state-of-the-art on challenging benchmarks without extra human prompts: it improves `gemma-2-9b-it`'s win-rate on Arena-Hard from 51.6% to 60.1% by DPO and 52.6% to 62.4% by RLOO, surpassing `claude-3-opus` and nearing `gemini-1.5-pro`, both are orders of magnitude larger. Further ablation studies show `eva` can induce meaningful learning curriculum, and effectively scale RL for LLMs beyond static human prompts.

## 1. Introduction

Long-lived artificial intelligence must deal with an ever-evolving, open-ended world, however the current training paradigm is restricted to being fairly short-lived and static.

This paper concerns the post-training paradigm of large language models (LLMs), which is typically done in two stages, imitation (*i.e.*, supervised fine-tuning, SFT) and reinforcement learning (*i.e.*, RL post-training). In particular, we focus on the latter, which has led to remarkable success

[1]Google DeepMind [2]University of Chicago. Correspondence to: Ziyu Ye and Yuan Liu <{hazelye, yuanliu}@google.com>. Approved for publication by Google DeepMind as of 2024-09-27.

---

**The Prioritized Prompt Evolving Principle**

LLMs shall adaptively control training prompts by *prioritizing* and *generating* useful prompts during RL.

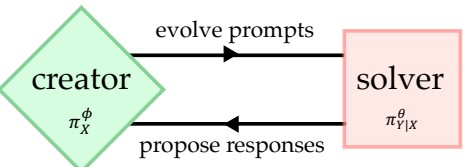

Figure 1: **RL for LLMs as Asymmetric Self-Play.** Conventional training methods are restricted to pre-curated prompt sets with fixed training schedules. We show that it is crucial to adaptively control the prompt distribution by creating and prioritizing more useful prompts. This is implemented through a simple self-play scheme ( Algorithm 1), where the **creator** is the newly introduced prompt generation policy $\pi_X$ and the **solver** is the response generation policy $\pi_{Y|X}$; both interact with objectives driven by reward signals (§ 3).

in enhancing LLM capabilities (Team et al., 2023; 2024a; 2025). However, there is a fundamental issue in existing practices: they restrict themselves within a *pre-curated*, *static* prompt distribution during post-training.

This is sub-optimal and bottlenecks scaling properties *w.r.t.*: (i) **training efficiency**: the existing paradigm treats all prompts equally, despite their utility depending on the changing states of LLMs in training; as not all prompts contribute equally, relying on a static set with a fixed training schedule is inefficient. (ii) **model generalizability**: once the LLM saturates on the static prompt set, learning stops, preventing the model acquiring new skills or knowledge beyond the predefined distribution.[1]

We thereby investigate the two questions:

1. (Signal) *Which prompts should be prioritized during RL training?*

2. (Algorithm) *How can we generate more useful prompts, and use them to keep LLMs self-improving in RL?*

---

[1]Literature has considered prompt evolving, but is restricted to SFT (Xu et al., 2023), and/or evolves uniformly (Yuan et al., 2024). We claim **adaptive training** *w.r.t.* **reward signals** are crucial.

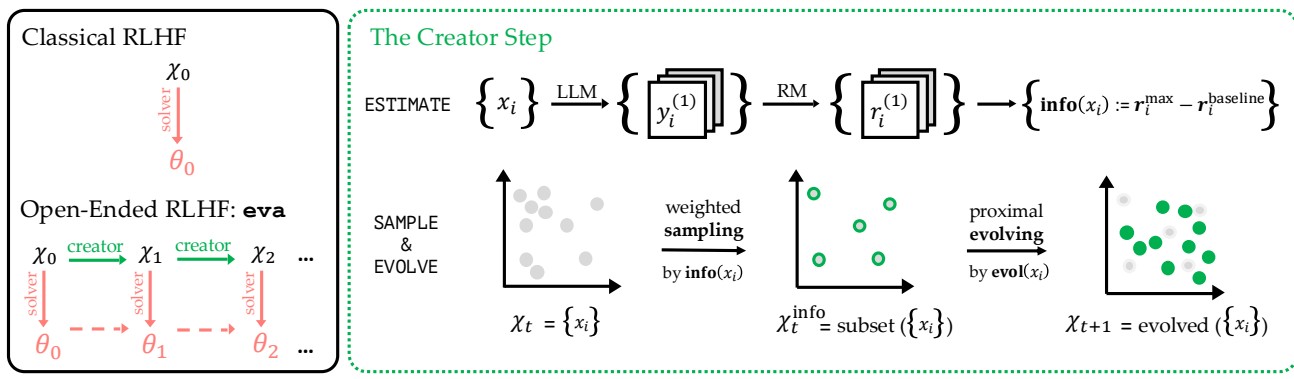

Figure 2: (Left) **eva** generalizes classical RL with open-ended RL for LLMs, by introducing a creator policy in training. (Right) The **creator** strategically controls prompt distributions by generating new prompts after each mini-batch gradient update (for online RL) or a complete dataset iteration (for offline RL), with a simple *estimate, sample then evolve* procedure, where the usefulness of each prompt is estimated from reward signals. The **solver** is then trained on a mixture of the original and evolved prompts. See the **minimax-regret** objective driving the game design and practical implementations in § 3.

To address them, we design **eva** (Evolving *via* Asymmetric Self-Play), as in Figure 1, 2. Central to **eva** is an infinite game with minimax-regret objectives, achieved by alternating optimization in creating prompts and solving them. We detail the novelty of this work *w.r.t.* prior works in § 5, and summarize our original contributions as:

1. **Signal**: We find **reward advantage** as an effective signal to identify useful prompts in RL post-training.

2. **Algorithm**: We design **eva**, the first method that enables LLMs to **prioritize** *and* **adaptively create** useful prompts, for continual RL training and self-improving.

The rest of technical details are organized as follows: in Problem 1, we formalize the problem of RL post-training as a bilevel optimization over a creator policy and a solver policy, which corresponds to the Stackelberg game (von Stackelberg, 1934) in game theory. In § 3.2, we define the objectives for the two players of the game by regret, for which the solver minimizes and the creator maximizes, and use its approximation by reward advantage for creator optimization. In § 3.3, we discuss simple practical algorithms of **eva** in both online and offline settings. In § 4, we run extensive experiments showing **eva** universally improves the performance of both online RL (*e.g.*, RLOO, OAIF) and offline RL (*e.g.*, DPO, SPPO, SimPO, ORPO) for LLMs and is SOTA on various challenging real-world benchmarks.

As we enter the new epoch where compute is moving from training to data synthesis, the question on "*how to put more compute in generating better data*" (Nishihara, 2025) is more critical than ever. While existing works (Zelikman et al., 2022; Gulcehre et al., 2023; Chow et al., 2024) primarily focus on the **exploration in** $\mathcal{Y} \mid \mathcal{X}$, we perform the first systematic study in **exploration in** $(\mathcal{X}, \mathcal{Y})$ in RL for LLMs, guided by carefully designed *reward signals*.

## 2. Preliminaries

Classical RL post-training (Ouyang et al., 2022) solves regularized optimization for a **static prompt dataset** $\mathcal{D}$:

$$\max_{\pi_{\boldsymbol{\theta}}} \mathbb{E}_{\mathbf{x} \sim \mathcal{D}, \mathbf{y} \sim \pi_{\boldsymbol{\theta}}(\cdot|\mathbf{x})} \Big[ r(\mathbf{x}, \mathbf{y}) \Big]$$
$$- \mathbb{E}_{\mathbf{x} \sim \mathcal{D}} \Big[ \beta \cdot \mathbb{D}_{\mathsf{KL}} \big[ \pi_{\boldsymbol{\theta}}(\cdot \mid \mathbf{x}) \parallel \pi_{\mathsf{base}}(\cdot \mid \mathbf{x}) \big] \Big].$$

Here, $\pi_{\mathsf{base}}(\cdot|\mathbf{x})$ is a base policy, $\mathbf{x}$ and $\mathbf{y}$ are prompts and responses, $\mathbb{D}$ is a divergence measure. To approximate the expectation over $\mathbf{y}$, we employ Monte Carlo sampling by generating $k$ responses per prompt, $\boldsymbol{y}_{(1)}, \ldots, \boldsymbol{y}_{(k)} \sim \pi_{\boldsymbol{\theta}}(\cdot \mid \mathbf{x})$ (Sutton et al., 1998; Ahmadian et al., 2024). In this paper, $r(\cdot)$ is the reward assumed to be from an oracle $r^{\star}(\cdot)$ and is fixed during post-training (Team et al., 2024a); we use the conventional term RLHF (reinforcement learning from human feedback) interchangeably with RL (post-)training, and focus on human preference optimization, *i.e.*, AI alignment (Leike et al., 2018), in our experiments. Nevertheless, the pipeline is compatible with any other reward types.

Conventionally, prompts $\mathbf{x}$ are scheduled in training either sequentially or *via i.i.d.* uniform sampling. Prior works explored active selection (Kawaguchi & Lu, 2020; Muldrew et al., 2024) or prioritized sampling (Schaul et al., 2015; Lee et al., 2022), but focus solely on existing data. In contrast, **eva** introduces new prompt creation, enabling training with improved coverage and complexity beyond the initial static prompt set. Depending on how $\mathbf{y}$ is generated and labeled, RL post-training methods can be described as: (i) **online** on-policy, where $\mathbf{y}$ are generated on-policy (Team et al., 2024a); and (ii) **offline**, where $\mathbf{y}$ are pre-generated by existing model checkpoints (Xiong et al., 2024; Pang et al., 2024a). In this paper, for online RLHF, we evolve prompts at each mini-batch, while for offline RLHF, we construct a new prompt set after each full iteration over the previous set.

---

**Problem 1 (Open-Ended RLHF)** *We define the problem of* Open-Ended RLHF *as the bilevel optimization on both a prompt policy (the creator $\pi_\phi(\mathbf{x})$) and a response policy (the solver $\pi_\theta(\mathbf{y} \mid \mathbf{x})$):*

$$\phi^\star \in \arg\max_{\phi} \ \mathcal{R}\Big( \pi_\phi(\cdot), \pi_{\mathsf{true}}(\cdot); \mathcal{D}, \theta^\star(\phi) \Big), \tag{1}$$

$$s.t. \quad \theta^\star(\phi) \in \arg\max_{\theta} \ \mathbb{E}_{\mathbf{x} \sim \pi_\phi(\cdot)}\Big[ \mathbb{E}_{\mathbf{y} \sim \pi_\theta(\cdot|\mathbf{x})}\big[ r(\mathbf{x}, \mathbf{y}) \big] - \beta \cdot \mathbb{D}_{\mathsf{KL}}\big[ \pi_\theta(\cdot \mid \mathbf{x}) \parallel \pi_{\mathsf{base}}(\cdot \mid \mathbf{x}) \big] \Big]. \tag{2}$$

*where $\pi_{\mathsf{true}}$ is the (potentially unknown) true target prompt distribution, $\mathcal{D}$ is an optional artifact parameter (e.g., the seed prompt distribution), and $R(\cdot)$ is a regularization function on the creator policy $\pi_\phi$, which we discuss in detail at § 3. The problem generalizes classical RLHF, and captures the dual objective that (i)* **response alignment***: the solver should perform well on the training prompt distribution while staying close to $\pi_{\mathsf{base}}$, and (ii)* **prompt generation***: the creator should generate training prompts allowing the solver to perform robustly on target prompt distributions.*

## 3. Method

### 3.1. The Problem: Open-Ended RLHF

Classical RLHF, as in § 2, samples prompts from a *static* set $\mathcal{D}$, which can have limited prompt coverage and complexity (Fig. 11), and may diverge from true scenarios in the *open-ended world* (Dennis et al., 2020; Parker-Holder et al., 2022). In Problem 1, we introduce a prompt generation policy $\pi_\phi(\cdot)$ to be optimized together with the response generation policy $\pi_\theta(\cdot|\mathbf{x})$. An optimizable $\pi_\phi(\cdot)$ brings a few benefits: (i) during training, it allows dynamic adjustment of training prompts on-the-fly, making it possible to prioritize prompts that are more informative to the current $\pi_\theta(\cdot|\mathbf{x})$, improving learning efficiency; (ii) at convergence, it brings a new prompt distribution beyond the initial static set, making it possible for $\pi_\theta(\cdot|\mathbf{x})$ to learn knowledge beyond $\mathcal{D}$ and perform more robustly on the target distribution. The creator objective $\mathcal{R}(\cdot)$ characterizes the optimization of $\pi_\phi(\cdot)$, preventing it from collapsing in trivial prompts and guiding it towards true target prompts, which we discuss a specific implementation by regret maximization in the next.

### 3.2. The Game: Minimax Regret Games

Problem 1 can be cast as a sequential game (von Stackelberg, 1934) by two strategic players optimizing each's utility:

- **Solver** $\pi_\theta(\mathbf{y} \mid \mathbf{x})$, who generates responses that optimize alignment given training prompts.
- **Creator** $\pi_\phi(\mathbf{x})$, who generates training prompts for the solver to perform well in the real world, knowing the solver will optimize over the generated.

A natural objective for the creator is to improve the solver's transfer performance (Bengio, 2012) on the true target prompt distribution $\pi_{\mathsf{true}}$: the closer $\pi_\phi$ is to $\pi_{\mathsf{true}}$, the better the solver performance is expected, thereby the higher the utility the creator may receive. If $\pi_{\mathsf{true}}$ is known, $\mathcal{R}(\cdot)$ can be instantiated by a $f$-divergence measure for distribution matching with $\pi_{\mathsf{true}}$. This work considers the case

when $\pi_{\mathsf{true}}$ is unknown a priori; the optimization then falls into a standard *decision under ignorance problem* (Savage, 1951; Gustafsson, 2022). Several decision rules can be considered, *e.g.*, randomization, that chooses training prompt distribution uniformly (Jiang, 2023); in this paper, we study *Minimax Regret Rule*, which finds a training distribution that minimizes solver's worst-case regret over all possible distributions (see also § 4.2.1). We abuse the notation $r$ to denote the reward with KL penalty, and **Regret** is defined as reward differences of $\pi_\theta$ and optimal policy $\pi_\theta^\star$:

$$\mathrm{Regret}(\pi_\phi, \pi_\theta) = \mathbb{E}_{\mathbf{x} \sim \pi_\phi(\cdot)}\Big[ \mathbb{E}_{\mathbf{y} \sim \pi_\theta^\star(\cdot|\mathbf{x})}\big[ r(\mathbf{x}, \mathbf{y}) \big]$$
$$- \mathbb{E}_{\mathbf{y} \sim \pi_\theta(\cdot|\mathbf{x})}\big[ r(\mathbf{x}, \mathbf{y}) \big] \Big].$$

Problem 1 is then formulated as:

$$\phi^\star \in \arg\max_{\phi} \ \mathrm{Regret}(\pi_\phi, \pi_{\theta^\star}), \tag{3}$$

$$s.t. \quad \theta^\star(\phi) \in \arg\min_{\theta} \ \mathrm{Regret}(\pi_\phi, \pi_\theta). \tag{4}$$

Note that (i) for solver optimization, Eq. 4 is equivalent to Eq. 2 by definition, and (ii) for creator optimization, Eq. 3 approximates Eq. 1 with a worst-case optimal guarantee for the solver's policy, when $\pi_{\mathsf{true}}$ is unknown.

**Remark 1** *Under mild assumptions, the (local) Nash equilibrium is a (local) minimax point (Jin et al., 2020) for the above optimization; here, at the (local) Nash equilibrium, the solver follows a (local) minimax regret policy (Jiang, 2023), i.e., the solver's regret is worst-case optimal.*

The equilibrium finding of the game can be solved by alternating optimization (Zhang et al., 2022). Intuitively, this allows for the creation of evolving prompt distributions that challenge the agent progressively for better generalization; the regret objective ensures *robustness* on such evolving curricula by *incentivizing agents to perform well in all cases*, providing a worst-case guarantee. In optimization, this brings a sweet spot where the creator can create challenging yet solvable prompts (*i.e.*, neither too hard nor too easy) for the solver, as illustrated in § H.3.

**Regret Minimization for the Solver.** Any preference optimization algorithms can be used as a plug-in for the regret minimization for the solver's step in Algorithm 1.

**Regret Maximization for the Creator.** When it is direct for the solver to minimize the regret by policy optimization, the true optimal policy remains unknown during optimization, and we must approximate it when using it as the utility to incentivize the creator. Similar to heuristics in prior works (Jiang et al., 2021b;a; Parker-Holder et al., 2022), we use the advantage-based estimate for each $\mathbf{x}$:

$$\hat{\text{Regret}}(\mathbf{x}, \pi_{\boldsymbol{\theta}}) \leftarrow r(\mathbf{x}, \mathbf{y}_+) - r(\mathbf{x}, \mathbf{y}_{\text{baseline}}), \quad (5)$$

where

$$\mathbf{y}_+ := \arg\max_{\mathbf{y}_i} r(\mathbf{x}, \mathbf{y}),$$

$$\mathbf{y}_{\text{baseline}} := \text{avg}_{\mathbf{y}_i} r(\mathbf{x}, \mathbf{y}) \text{ or } \arg\min_{\mathbf{y}_i} r(\mathbf{x}, \mathbf{y}),$$

and $\{\mathbf{y}_i\}_{i=1}$ is a set of responses sampled from $\pi_{\boldsymbol{\theta}}(\cdot \mid \mathbf{x})$ and $r(\cdot, \cdot)$ is the reward oracle. We choose $\arg\text{avg}_{\mathbf{y}_i} r(\mathbf{x}, \mathbf{y})$ for online RLHF, and $\arg\min_{\mathbf{y}_i} r(\mathbf{x}, \mathbf{y})$ for offline RLHF, based on consistent strong empirical gains observed across extensive experiments. As the policy optimizes, the proxy will approximate the true regret better[2]. We denote the regret estimate (*i.e.*, negated reward advantage) as the informativeness value for a prompt $\mathbf{x}$ *w.r.t.* $\boldsymbol{\theta}$,

$$\text{info}_{\boldsymbol{\theta}}(\mathbf{x}) := \hat{\text{Regret}}(\mathbf{x}, \pi_{\boldsymbol{\theta}}). \quad (6)$$

Directly doing gradient ascent on regret could lead to training instability (Zhang, 2023). In this work, we *approximate new prompt distributions* that maximize regret by 3 steps:

1. **Estimate informativeness** for each prompt in the set.

2. **Sampling a subset of high-regret prompts**.

3. **Generating new prompts** by making variations on those high-regret prompts.

The **eva** way of scalable regret maximization can relate to curriculum RL (Parker-Holder et al., 2022), which finds environments with high-regret levels, then edits within some distance, or *evolution strategies* (Schwefel, 1977) which find the most promising species, then mutate and crossover.

### 3.3. The Practical Algorithm

Algo 1 is an overview of **eva** where the creator constructs new training prompts after a full iteration of the solver.

#### 3.3.1. THE SOLVER STEP

This step is the classical preference optimization (Rafailov et al., 2023). Take DPO as an example, for every prompt,

---

[2]This approximates the expectation over $\pi_{\boldsymbol{\theta}}^{\star}$ by best observed responses, which introduces bias quantifiable by $\mathbb{D}_{\text{KL}}[\pi_{\boldsymbol{\theta}} \| \pi^{\star}]$. Also, single-sample estimation for $\pi_{\phi}$ introduces variance. See Liu et al. (2023a) for discussions on sampling effects.

---

**Algorithm 1 eva.**

    **Input:** initial policy $\pi_{\boldsymbol{\theta}_0}$, initial set of prompt $\mathcal{X}_0$
1: **for** iteration $t = 1, 2, \ldots$ **do**
      $\triangledown$   /* **creator step** */
2:     *estimate*:     $\mathcal{X}_{t-1} \leftarrow \{(\boldsymbol{x}_i, \text{info}(\boldsymbol{x}_i)) \mid \boldsymbol{x}_i \in \mathcal{X}_{t-1}\}$
     *sample*:     $\mathcal{X}_{t-1}^{\text{info}} \leftarrow \{\boldsymbol{x}_i \text{ drawn w.p.} \propto \text{info}(\boldsymbol{x}_i)\}$
     *evolve*:     $\mathcal{X}_t \leftarrow \text{evolve}(\mathcal{X}_{t-1}^{\text{info}})$

      $\triangledown$   /* **solver step** */
3:     *generate*:     $\forall \boldsymbol{x}_i \in \mathcal{X}_t^{\text{info}}, \{\boldsymbol{y}_i^{(j)}\} \sim \pi_{\boldsymbol{\theta}_{t-1}}(\cdot \mid \boldsymbol{x}_i)$
     *annotate reward*:   $\mathcal{X}_t' \leftarrow \mathcal{X}_t^{\text{info}} \cup \{(\boldsymbol{y}_i^{(j)}, r_i^{(j)})\}$
     *optimization*:   $\boldsymbol{\theta}_t \leftarrow \boldsymbol{\theta}_{t-1} + \eta \nabla_{\boldsymbol{\theta}} \mathcal{J}_{\mathcal{X}_t'}(\boldsymbol{\theta})$
4: **end for**
5: **return** final solver policy $\pi_{\boldsymbol{\theta}_T}$

---

we sample $n$ responses and annotate rewards, then take the responses with the maximal and the minimal reward to construct preference pairs and optimize upon.

#### 3.3.2. THE CREATOR STEP

Plainly, the creator finds most useful prompts and generate variants of them to approximate regret maximization.

**Step 1: info($\cdot$) –** *estimate the informativeness.* For each $\mathbf{x}$ in the prompt set $\mathcal{X}_t$, we generate responses, annotate rewards and estimate the informativeness of $\mathbf{x}$ by Eq. 6.

**Step 2: sample($\cdot$) –** *weighted sampling for an informative subset.* By using the informativeness metric as the weight, we sample an informative subset $\mathcal{X}_t^{\text{info}}$ to be evolved.

**Step 3: evolve($\cdot$) –** *evolving for high-regret prompts.* **eva** is agnostic to and does not rely on any specific evolving method (see empirical evidence in § H.1). We take Xu et al. (2023) as a default baseline for offline RLHF, with in-depth and in-breadth instructions for prompt re-writing.

As a side note, we discuss a useful technique below.

**Prioritized Generative Buffer.** While **eva** can operate on the full $\mathcal{D}$ at once and iteratively train LLMs (*i.e.*, offline **eva**), the informativeness can become off-policy. Inspired by Schaul et al. (2015), we design a simple *prioritized generative buffer* $\mathcal{B}$ that extends Algorithm 1 to be on-policy and evolves per mini-batch (*i.e.*, online **eva**), with:

1. **Warm-up phase**: we start with $\mathbf{x}$ from $\mathcal{D}$ and populate $\mathcal{B}$ with evolved prompts until it reaches size $B$.

2. **Mix-up phase**: training continues using a balanced mix of samples from $\mathcal{D}$ and $\mathcal{B}$ (prioritized by informativeness) per mini-batch. New prompts are evolved and added to $\mathcal{B}$, while older ones are removed.

3. **Bootstrap phase**: Once $\mathcal{D}$ is exhausted, training relies on $\mathcal{B}$, continuing evolving and replacing prompts.

**eva** is easy to use and flexible to extend. We provide detailed instructions for practitioners in the Appendix.

# 4. Experiments

**Datasets and models.** We use **UltraFeedback** (Cui et al., 2023) as the training dataset, which contains diverse high-quality prompts that are primarily human-generated. We use the instruction-finetuned GEMMA-2-9B (Team et al., 2024b) as the base ($\theta_0$)[3], which is a strong baseline for models of its size. Note that we directly apply RL training without SFT, as the base model is sufficiently capable.

**Evaluation settings.** We use: (i) **AlpacaEval 2.0** (Dubois et al., 2024), which assesses general instruction following with 805 questions; (ii) **MT-Bench** (Zheng et al., 2023), which evaluates multi-turn instruction following with 80 hard questions in 8 categories; (iii) **Arena-Hard** (Li et al., 2024), which is derived from 200K user queries on Chatbot Arena with 500 challenging prompts across 250 topics.

**Optimization algorithms.** We evaluate our method across a wide range of six representative RLHF algorithms:

- **Online RLHF**: RLOO (Ahmadian et al., 2024), OAIF (*i.e.*, online DPO) (Guo et al., 2024).
- **Offline RLHF**: (*with reference*) DPO (Rafailov et al., 2023), SPPO (Wu et al., 2024); (*without reference*) SimPO (Meng et al., 2024), ORPO (Hong et al., 2024).

**Oracle reward models.** We take ARMORM-8B (Wang et al., 2024) to be the default reward model for human-preference proxy, with the below for ablation studies:

- **Pointwise**: ARMORM-8B (Wang et al., 2024), SKYWORKRM-27B (Liu & Zeng, 2024).
- **Pairwise**: PAIRRM-0.4B (Jiang et al., 2023), PAIRRM-8B (Dong et al., 2024).

## 4.1. Main Results

**eva consistently achieves strong self-improvement.** As in in Table 1 and 2, **eva** yields notable performance improvement across different optimization algorithms, especially on the more challenging and robust Arena-Hard benchmark (Li et al., 2024). For example, **eva** brings $10.6\%$ gain with DPO in the offline setting, and $9.8\%$ gain with RLOO in the online setting, surpassing `claude-3-opus-240229` as reported by AH leaderboard and matching `gemini-1.5-pro`, while using fully adaptive self-automated joint prompt-response generation. This demonstrates the superior empirical performance of **eva**.

---

[3]Unless stated otherwise, each iteration uses 10K prompts (the initial prompt set), referred to as 1x. In offline RLHF, we denote $\theta_{t \to t+1}$ as the one trained with new human prompts from the $t$-th checkpoint. $\theta_{t \to \tilde{t}}$ denotes the one trained with evolved prompts from the $t$-th checkpoint without any new human prompts. In online RLHF, training is a continual iteration and $\theta_{0 \to \tilde{1}}$ ($n$x) denotes training with 10$n$K prompts in total, mixed and evolved from the initial.

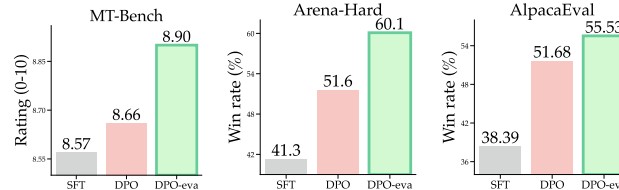

Figure 3: Illustration of gains with one round **eva** by DPO.

Table 1: **Online eva results. eva** has notable gains and is comparable to default training with even 6x human prompts (gray). Note **eva only uses** 1x **human prompts** and continuously evolves ($n$x denotes total prompt size).

| Optimization Method ($\to$) | Online RLHF | | | |
|---|---|---|---|---|
| Benchmark ($\to$) | **Arena-Hard** | **MT-Bench** | | **AE 2.0** |
| Method ($\downarrow$) / Metric ($\to$) | WR (%) | avg. | turn 1 | turn 2 | LC-WR (%) |
| $\theta_0$: Base Model | 41.3 | 8.57 | 8.81 | 8.32 | 47.11 |
| $\theta_{0\to1}$: RLOO (1x) | 52.6 | 8.68 | 9.02 | 8.34 | 54.23 |
| $\theta_{0\to\tilde{1}}$: RLOO-**eva** (1x) | **57.3** | 8.87 | 9.03 | 8.71 | **55.02** |
| $\theta_{0\to\tilde{1}}$: RLOO-**eva** (2x) | **60.5** | 8.96 | 9.12 | 8.80 | **57.10** |
| $\theta_{0\to\tilde{1}}$: RLOO-**eva** (3x) | **62.4** | 9.09 | 9.23 | 8.94 | **61.04** |
| $\theta_{0\to1}$: RLOO (6x) | 62.7 | 9.07 | 9.24 | 8.90 | 62.91 |
| $\theta_{0\to1}$: OAIF (1x) | 52.1 | 8.66 | 8.97 | 8.35 | 55.15 |
| $\theta_{0\to\tilde{1}}$: OAIF-**eva** (1x) | **55.0** | 8.85 | 9.04 | 8.66 | **55.43** |
| $\theta_{0\to\tilde{1}}$: OAIF-**eva** (2x) | **60.4** | 8.93 | 9.06 | 8.79 | **56.49** |
| $\theta_{0\to\tilde{1}}$: OAIF-**eva** (3x) | **61.7** | 9.01 | 9.19 | 8.82 | **59.09** |

Table 2: **Offline eva results.** We apply **eva** after 1 iteration of offline RLHF. It brings strong gains and can surpass training with human prompts. See more iterations in § 4.2.4.

| Optimization Method ($\to$) | Offline RLHF | | | |
|---|---|---|---|---|
| Benchmark ($\to$) | **Arena-Hard** | **MT-Bench** | | **AE 2.0** |
| Method ($\downarrow$) / Metric ($\to$) | WR (%) | avg. | turn 1 | turn 2 | LC-WR (%) |
| $\theta_0$: Base Model | 41.3 | 8.57 | 8.81 | 8.32 | 47.11 |
| $\theta_{0\to1}$: DPO | 51.6 | 8.66 | 9.01 | 8.32 | 55.01 |
| $\theta_{1\to\tilde{1}}$: + **eva** | **60.1** | **8.90** | **9.04** | **8.75** | 55.35 |
| $\theta_{1\to2}$: + new human prompts | 59.8 | 8.64 | 8.88 | 8.39 | 55.74 |
| $\theta_{0\to1}$: SPPO | 55.7 | 8.62 | 9.03 | 8.21 | 51.58 |
| $\theta_{1\to\tilde{1}}$: + **eva** | **58.9** | **8.78** | **9.11** | **8.45** | **51.86** |
| $\theta_{1\to2}$: + new human prompts | 57.7 | 8.64 | 8.90 | 8.39 | 51.78 |
| $\theta_{0\to1}$: SimPO | 52.3 | 8.69 | 9.03 | 8.35 | 54.29 |
| $\theta_{1\to\tilde{1}}$: + **eva** | **60.7** | **8.92** | **9.08** | **8.77** | **55.85** |
| $\theta_{1\to2}$: + new human prompts | 54.6 | 8.76 | 9.00 | 8.52 | 54.40 |
| $\theta_{0\to1}$: ORPO | 54.8 | 8.67 | 9.04 | 8.30 | 52.17 |
| $\theta_{1\to\tilde{1}}$: + **eva** | **60.3** | **8.89** | **9.07** | **8.71** | **54.39** |
| $\theta_{1\to2}$: + new human prompts | 57.2 | 8.74 | 9.01 | 8.47 | 54.00 |

**eva curricula can surpass human-crafted prompts.** We further show that **eva** models can match and even outperform those trained on additional new prompts from Ultra-Feedback (denoted as new human prompts as they are primarily sourced from humans (Cui et al., 2023)), while being much more efficient. Interestingly, on MT-Bench, training with new human prompts typically show decreased performance in the 1st turn and only modest gains in the 2nd turn, whereas **eva** notably enhances 2nd gains. We hypothesize that **eva** adaptively evolves novel, learnable prompts that include features of second-turn questions, reflecting generalized skills like handling follow-up interactions.

## 4.2. Ablation Studies

Taking offline DPO as a representative case, we conduct extensive ablation studies on **eva**, with key findings:

- § 4.2.1 - **informativeness metric**: our *regret*-based metric outperforms other alternatives.

- § 4.2.2 - **adaptive evolving procedure**: our method outperforms active selection without evolving.

- § 4.2.3 - **scaling with reward models**: the alignment gain of **eva** scales with reward models.

- § 4.2.4 - **continual training** : our method has monotonic gain with incremental training.

- § 4.2.5 - **curriculum effect**: our method creates meaningful curriculum over iterations.

### 4.2.1. THE CHOICE OF INFORMATIVENESS METRICS

| Metric | info($\mathbf{x}$) | Related Approximation |
|---|---|---|
| $A^\star_{\min}$: worst-case optimal advantage | $|\min_{\mathbf{y}} r(\mathbf{x}, \mathbf{y}) - \max_{\mathbf{y}'} r(\mathbf{x}, \mathbf{y}')|$ | minimax regret (Savage, 1951) |
| $A^\star_{\text{avg}}$: average optimal advantage | $|\frac{1}{N}\sum_{\mathbf{y}} r(\mathbf{x}, \mathbf{y}) - \max_{\mathbf{y}'} r(\mathbf{x}, \mathbf{y}')|$ | Bayesian regret (Banos, 1968) |
| $A^\star_{\text{dts}}$: dueling optimal advantage | $|\max_{\mathbf{y} \neq \mathbf{y}^\star} r(\mathbf{x}, \mathbf{y}) - \max_{\mathbf{y}'} r(\mathbf{x}, \mathbf{y}')|$ | min-margin regret (Wu & Liu, 2016) |

Table 3: The reward-advantage-based metrics that serve as the informativeness proxies for prompts.

| Benchmark ($\rightarrow$) | Arena-Hard | MT-Bench | | | AE 2.0 |
|---|---|---|---|---|---|
| Method ($\downarrow$) / Metric ($\rightarrow$) | WR (%) | avg. | turn 1 | turn 2 | LC-WR (%) |
| $\theta_{0\rightarrow1}$: DPO | 51.6 | 8.66 | 9.01 | 8.32 | 55.01 |
| $\theta_{1\rightarrow\tilde{1}}$:  + **eva** (uniform) | 57.5 | 8.71 | 9.02 | 8.40 | 53.43 |
| $\theta_{1\rightarrow\tilde{1}}$:  + **eva** (var($\boldsymbol{r}$)) | 54.8 | 8.66 | 9.13 | 8.20 | 54.58 |
| $\theta_{1\rightarrow\tilde{1}}$:  + **eva** (avg($\boldsymbol{r}$)) | 58.5 | 8.76 | 9.13 | 8.40 | 55.01 |
| $\theta_{1\rightarrow\tilde{1}}$:  + **eva** (1/avg($\boldsymbol{r}$)) | 56.7 | 8.79 | 9.13 | 8.45 | 55.04 |
| $\theta_{1\rightarrow\tilde{1}}$:  + **eva** (1/$A^\star_{\min}$) | 52.3 | 8.64 | 8.96 | 8.31 | 53.84 |
| $\theta_{1\rightarrow\tilde{1}}$:  + **eva** ($A^\star_{\text{avg}}$) (our variant) | 60.0 | 8.85 | 9.08 | 8.61 | **56.01** |
| $\theta_{1\rightarrow\tilde{1}}$:  + **eva** ($A^\star_{\text{dts}}$) (our variant) | 60.0 | 8.86 | **9.18** | 8.52 | 55.96 |
| $\theta_{1\rightarrow\tilde{1}}$:  + **eva** ($A^\star_{\min}$) (our default) | **60.1** (+8.5) | **8.90** | 9.04 | **8.75** (+0.43) | 55.35 |

Table 4: **Choice of informativeness metric matters.** Our adaptive metric by *reward advantage* achieves the best performances. See also § D for visualization.

**Reward advantage as the informativeness metric outperforms baselines.** As in Table 4, **eva** offers an effective curriculum by the advantage-based proxy as the informativeness metric (bottom row):

- *Comparing with uniform evolving* (brown): Existing baselines generate prompts in a uniform manner (Yuan et al., 2024) (*cf.*, the principle of insufficient reason (Keynes, 1921; Tobin et al., 2017)). **eva** concretely outperforms, corroborating (Das et al., 2024) that uniform learners can suffer from sub-optimality gaps.

- *Comparing with other heuristics* (gray): Prior practices (Team et al., 2023) tried heuristics like prioritizing prompts with the most variance in its rewards or with the lowest/highest average. We find our advantage based methods (red) outperforms those heuristics.

- *Comparing with the inverse advantage* (purple): Contrary to curriculum learning, a line of works conjecture that examples with higher losses may be prioritized (Jiang et al., 2019), which can be done by inverting our metric. We find it significantly *hurt* the alignment gain, corroborating (Mindermann et al., 2022) that those examples can be unlearnable or irrelevant, meaning our curriculum is effective and practical.

- *Among our advantage variants* (green): We designed variants of our default advantage-based metric, as in Table 3; the default $A^\star_{\min}$ remains competitive among its peers. Together, the advantage-based principle provides a robust guideline for sampling and evolving.

The lesson is that we must be selective about which are the promising to evolve, otherwise unlearnable, noisy or naïve prompts may hinder learning. Our regret-inspired metric represents a solid baseline.

### 4.2.2. THE EFFECT OF EVOLVING

| Benchmark ($\rightarrow$) | Arena-Hard | MT-Bench | | | AlpacaEval 2.0 | |
|---|---|---|---|---|---|---|
| Method ($\downarrow$) / Metric ($\rightarrow$) | WR (%) | avg. | turn 1 | turn 2 | LC-WR (%) | WR (%) |
| $\theta_{0\rightarrow1}$: DPO | 51.6 | 8.66 | 9.01 | 8.32 | 55.01 | 51.68 |
| $\theta_{1\rightarrow\tilde{1}}$:  [no evolve]-greedy | 56.1 | 8.68 | 8.98 | 8.38 | 54.11 | 53.66 |
| $\theta_{1\rightarrow\tilde{1}}$:  [no evolve]-sample | 55.3 | 8.69 | 9.00 | 8.38 | 54.22 | 54.16 |
| $\theta_{1\rightarrow\tilde{1}}$:  + **eva**-greedy (our variant) | 59.5 | 8.72 | 9.06 | 8.36 | 54.52 | 55.22 |
| $\theta_{1\rightarrow\tilde{1}}$:  + **eva**-sample (our default) | **60.1** | **8.90** | 9.04 | **8.75** | **55.35** | **55.53** |

Table 5: **Effect of evolving.** The blue are those training with only the informative subset and without evolving); we denote -sample for the default weighted sampling procedure in Algo 1, while using -greedy for the variant from the classical active data selection procedure (*cf.*, a recent work (Muldrew et al., 2024) and a pre-LLM work (Kawaguchi & Lu, 2020)), which selects data by a high-to-low ranking via the metric greedily. We show evolving brings a remarkable alignment gain (green v.s. blue); and as we evolve, sampling is more robust than being greedy.

**The design of eva(·) is effective.** As in Table 5:

- Removing the evolve(·) step: if we only do subset sampling or ordered selection, we still have gain, but not as much as with evolving (*e.g.*, **eva** brings $4.8\%$ additional wins on Arena Hard).

- Altering the sample(·) step: if we greedily select prompts by the metric instead of using them as weights for importance sampling, the performance will be weaker as we evolve.

The lesson is that simply adaptive training within a fixed prompt distribution is not enough; our open-ended RLHF with *generative* prompt exploration gives a substantial headroom for self-improvement. In other words, the RL post-training process should be both *adaptive* and *generative* in terms of prompt distribution.

### 4.2.3. SCALING eva WITH REWARD MODELS

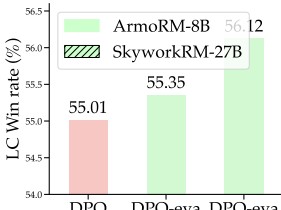 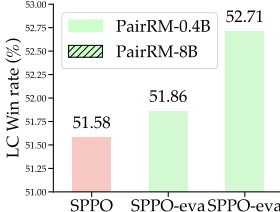

Figure 4: **eva scales with quality of reward models**, under pointwise RMs with DPO (*left*) and pairwise RMs with SPPO (*right*). Note SPPO handles general preferences thus requires pairwise RMs, and DPO relies on the Bradley-Terry assumption, for which pointwise RMs are suitable.

Figure 4 presents the length-controlled win rate of **eva** on AlpacaEval using pointwise and pairwise reward models of varying scales. As the quality of reward models improve, **eva** brings higher alignment gain. The scaling observation shows the effectiveness of **eva** in exploiting more accurate reward signals to choose informative prompts for better alignment. One takeaway is interaction with the external world is essential for intelligence. The more accurate reward signals observed, the better the agent incentivize themself to improve (*cf.*, (Silver et al., 2021)).

### 4.2.4. eva IMPROVES EFFICIENCY & GENERALIZATION

We run the default *incremental training* (*i.e.*, trainining from the last checkpoint with the evolved set in each iteration), as in Fig 5 and § H.2, **eva** presents *monotonic gains*.

The solutions found by **eva** *cannot* be recovered by training longer by a fixed set (the dashed), nor by naïvely sourcing new prompts without examining informativeness (the gray dotted), thus our generative data schedule is effective.

We conjecture that behaviors of the dashed/dotted lines relate to *loss of plasticity* (Ash & Adams, 2019; Dohare et al., 2023; Abbas et al., 2023; Xue et al., 2024). Classical works resolve it by the *optimization* view (*e.g.*, weight perturbing), whereas **eva** offers a new *data* view, potentially mimicking an **implicit regularizer for better generalization**.

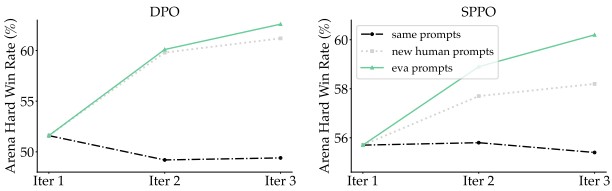

Figure 5: **eva** stays robust with more iterations.

In Table 6, we ablate **eva** in *scratch training*, *i.e.*, training with the full original *and* evolved set. **eva** is competitive in incremental training, *learning more effectively* with *less data* – a nice bonus by minimax regret (Jiang et al., 2021a).

Table 6: **Ablation on incremental v.s. scratch training.**

| Benchmark (→) | Arena-Hard | MT-Bench | AE 2.0 |
|---|---|---|---|
| Method (↓) / Metric (→) | WR (%) | avg. score | LC-WR (%) |
| $\theta_0$: SFT | 41.3 | 8.57 | 47.11 |
| $\theta_{0\to1}$: DPO | 51.6 | 8.66 | 55.01 |
| $\theta_{0\to\tilde{1}}$:  eva (scratch) | 59.8 | 8.88 | 54.59 |
| $\theta_{1\to\tilde{1}}$:  eva (incremental) | **60.1** | **8.90** | **55.35** |

### 4.2.5. eva CREATES MEANINGFUL CURRICULUM

Table 7: **eva** improves prompt quality and complexity.

| Prompt Set (↓) / Metric (→) | Complexity (1-5) | Quality (1-5) |
|---|---|---|
| UltraFeedback (seed) | 2.90 | 3.18 |
| UltraFeedback-**eva**-Iter-1 | 3.84 | 3.59 |
| UltraFeedback-**eva**-Iter-2 | 3.92 | 3.63 |
| UltraFeedback-**eva**-Iter-3 | **3.98** | **3.73** |

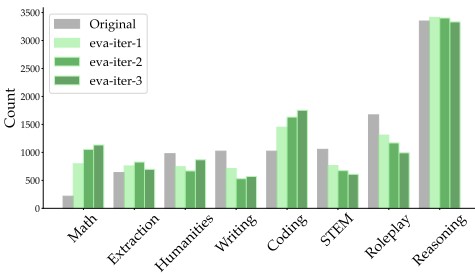

Figure 6: **Curriculum effect in training distributions.** The prompt distribution of Table 13. **eva** creates a curriculum that prioritizes math / coding prompts over iterations.

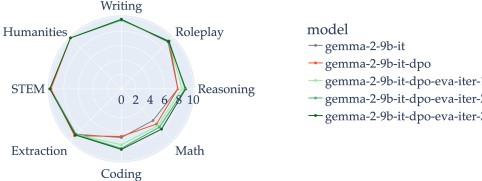

Figure 7: **Curriculum effect in benchmark performance.** The radar figure for ratings on MT-Bench. **eva** prioritizes and gradually improves on coding, math and reasoning over iterations, implicitly reflecting a learned curriculum.

In Table 7, we show that there is a gradual improvement in prompt complexity and quality[4] over iterations with **eva**. In Figure 6 and 7, we show that **eva** brings auto-curricula and the creator is incentivized to create new prompts that are informative *w.r.t.* the current solver policy.

Together, those evidences supports the importance of *adaptively evolving* prompts jointly with responses, which we believe to be crucial in scaling up next-gen RL post-training.

---

[4]We use `gemini-1.5` as the generative scorer (§ E).

# 5. Related Works

With the history of machine learning, it is not new that self-play and data exploration brings intelligence (*e.g.*, Schmidhuber (1991)). We believe, to our knowledge, **eva** is among the first empirical works that systematically studied adaptive prompt evolving in RL post-training on large-scale LLM benchmarks. Below, viewing from many different angles, we list **eva**'s distinct impact and contribution.

**Self-improving algorithms and iterative optimization.** This line of work focuses on iteratively generating samples from the response policy and continuously re-training the policy by selected self-generated samples. Major works include ReST (Gulcehre et al., 2023; Singh et al., 2023), STaR (Zelikman et al., 2022), RFT (Yuan et al., 2023), RAFT (Dong et al., 2023), self-improving LLMs (Huang et al., 2022; Yuan et al., 2024); in the context of preference optimization, iterative DPO (Tran et al., 2023; Xiong et al., 2024; Pang et al., 2024b) has proven effective. Most works focus on self-training by improving in $\mathcal{Y} \mid \mathcal{X}$, while we **jointly optimize** both responses and prompts via generative exploration in $(\mathcal{X}, \mathcal{Y})$, allowing for continual RLHF.

**Prompt synthesis for language models.** Major works include Self-Instruct (Wang et al., 2022), WizardLM (Xu et al., 2023; Luo et al., 2023), Self-Align (Sun et al., 2024), EvoPrompt (Guo et al., 2023), Magpie (Xu et al., 2024), etc. **eva** is orthogonal to them since any such method can be plugged in as the `evolve(·)` for the creator (note we contribute to a small trick with BoN tree in § 3). We focus on the **RL post-training** phase, while those works are primarily SFT. Importantly, our work proposes **an adaptive metric** to sample and evolve prompts – a secret sauce is that prompts shall be prioritized *w.r.t.* the *informativeness*, whereas prior works mostly evolve *uniformly* (Yuan et al., 2024). Furthermore, most works evolves in an *offline* manner, while **eva**, to our knowledge, is the first framework that supports **online/on-policy evolving** in general RL post-training.

**Active and curriculum learning.** This line of works re-order training examples for efficiency (Bengio et al., 2009; Mindermann et al., 2022; Kawaguchi & Lu, 2020), with recent LLM-related works (Muldrew et al., 2024; Das et al., 2024) (note those are done at a much smaller scale compared to **eva**). In contrast, **eva** breaks free from the static paradigm, not only re-orders data but also **generatively creates** new data, yielding significant gains (§ 4.2.2).

**Self-play and curriculum RL.** Agents trained on a fixed data distribution are often brittle and may struggle to adapt to the real world (Hughes et al., 2024). Self-play (Samuel, 1959; Goodfellow et al., 2014a; Silver et al., 2016) addresses this by having the agent learn through self-interaction, thus creating more diverse experiences and automatic curricula. In asymmetric self-play, the paradigm centers on "*Alice proposing a task, and Bob doing it*" (Sukhbaatar et al., 2017; Samvelyan et al., 2023; Beukman et al., 2024; Dennis et al., 2020). We revive the classical asymmetric self-play (Sutton et al., 2011) in optimizing language models. Unlike traditional curriculum RL (Parker-Holder et al., 2022), which renders environments by specifying levels (Dennis et al., 2020), our approach is *generative* by nature, as we directly **generate states** from powerful generative models.

**Self-play in RLHF.** A growing line of research frames RLHF as a *symmetric* self-play game, where both players are response players (Munos et al., 2023; Wu et al., 2024; Choi et al., 2024; Rosset et al., 2024). However, these methods still rely on a fixed prompt distribution thus is sub-optimal. In contrast, we solve this by **asymmetric** self-play, enabling evolving prompt distributions; this step is structurely resembles adversarial training (Goodfellow et al., 2014b; Ho & Ermon, 2016). (Zheng et al., 2024) is a concurrent work with similar asymmetric setup, however it applies to safety tasks instead of general instruction-following tasks and is incompatible with direct preference optimization.

# 6. Concluding Remarks

**Future directions.** **eva** defines a new paradigm for RL post-training, opening up many new directions, *e.g.*, (i) jointly optimizing the reward model (RMs) with **eva** – we assume a fixed oracle RM, as is *de facto* practice in industry (Team et al., 2024a); however, as the policy updates, **eva** can generate out-of-distribution prompts, necessitating the need for continual RM training (Makar-Limanov et al., 2024); (ii) extending to differentiable creator policies; (iii) extending to reasoning (Poesia et al., 2024); (iv) extending the game with more modality (Bruce et al., 2024), and/or with more players (*e.g.*, rewriters (Kumar et al., 2024)).

**Conclusions.** This empirical work presents **eva**, a new, simple and scalable framework for RL post-training of LLMs, that **adaptively generates** prompt distributions during training. **eva** is simple – it can be easily plugged into any existing pipeline, and highly effective – it reaches new SOTA on challenging alignment benchmarks. The primary takeaway may be: (i) self-evolving joint training distributions $(\mathcal{X}, \mathcal{Y})$ brings significant gain, and (ii) reward advantage acts as an effective metric informing the collection and creation of prompts in RLHF. Philosophically, **eva** presents a new view of post-training as an infinite game (Abel et al., 2024); **eva** incentivizes agents to create problems rather than to simply solve, which is key to intelligence, yet LLM trainers may neglect.

We believe the community should be aware of the surprising effectiveness of *adaptive evolving* prompts in RL post-training, and scale them together with responses.

## Impact Statement

`eva` enables scalable training of language agents through open-ended training, improving AI alignment with human values, which will ultimately contribute to social welfare (Pigou, 1920; Arrow, 1952; Rosenfeld & Xu, 2025). This may democratize the development of more generally capable artificial intelligence agents, impacting a wide range of domains from scientific discovery to societal governance.

We recognize that `eva` relies on self-exploration guided by reward signals. If these signals are inaccurate or misaligned, the trained agents may exhibit undesirable behaviors such as reinforcing biases or hallucinations. Mitigating these risks requires continued research into robust reward models, transparent evaluation protocols, open collaboration within the AI research community, and more. As authors, we will be committed to supporting these efforts by sharing our findings and implementations to promote open and responsible research and development.

## Acknowledgments

We extend our warm gratitude to Bilal Piot for his thoughtful reviews and valuable suggestions on this paper. We are also grateful to Chenkai Kuang, Dustin Tran, Albert Webson, Hanzhao Lin, Clara Huiyi Hu, Jeremiah Liu, Luheng He, Chenjie Gu, Yong Cheng, Pei Sun, and Heng-Tze Cheng for their fun discussions and interesting notes on Self-Play. We thank David Abel, Yuxin Chen, Minbiao Han, Feiran Jia, Ziniu Hu, Guohao Li, Rylan Schaeffer, Haifeng Xu, Chaoqi Wang and Yifei Wang for their helpful discussions and references on fine-tuning, contrastive learning, data synthesis, open-ended learning, and continual reinforcement learning. We also thank many anonymous reviewers.

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

# A. Details on Reproducibility

**Software environments.** All experiments are conducted on 8xNVIDIA H100 SXM GPUs. Our codebase primarily relies on `transformers==4.40.0`. For the response generation of GEMMA models at the training stage, we use `vllm==0.5.4` with `flashinfer` backend for CUDA 12.4 and `torch 2.4`. For evolving prompts, we use `distilabel==1.3.2`, and use `LiteLLM` to serve Gemini (default to be `gemini-1.5-pro`) and transformers models (default to be `gemma-2-9b-it`). For evaluation on all benchmarks, we use `sglang==0.2.10` and `openai==1.35.14`, with `gpt-4-1106-preview` as the judge model and `gpt-4-0314-preview` as the baseline model. Specifically for AlpacaEval 2.0, we use `alpaca_eval_gpt4_turbo_fn` as the annotator config.

**Hyperparameter settings.** We follow the original hyperparameter settings as in (Hong et al., 2024; Meng et al., 2024; Wu et al., 2024), default to be:

| Hyperparameter ($\downarrow$) / Loss ($\rightarrow$) | DPO | ORPO | SimPO | SPPO |
|---|---|---|---|---|
| learning rate | 5e-7 | 5e-7 | 8e-7 | 5e-7 |
| learning rate scheduler | cosine | cosine | cosine | linear |
| $\beta$ | 0.05 | / | 10 | 0.001 |
| $\gamma$ | / | / | 5 | / |
| $\lambda$ | / | 0.5 | / | / |
| no. epochs per iter | 2 | 1 | 1 | 6 |
| warmup ratio per iter | 0.1 | 0.1 | 0.1 | 0.1 |
| effective batch size | 8 | 8 | 32 | 8 |
| max length | 2048 | 2048 | 2048 | 1024 |
| max prompt length | 1024 | 1024 | 1024 | 512 |
| optimizer | adamw | adamw | adamw | rmsprop |

**Iterative Training Settings for Offline `eva`.** By default, we train with equal-size prompt subset in each iteration. Unless otherwise specified, we use 10K prompts from the UltraFeedback dataset (Cui et al., 2023) per iteration. The incremental training proceeds as follows (note this is also compatible with online solvers):

- $\theta_0$ : Base SFT model.

- $\theta_{0\rightarrow1}$ : initialize with $\theta_0$; then train w/ the prompt split $\mathcal{X}_1$ by self-generated responses from the initial model $\theta_0$.

- $\theta_{1\rightarrow2}$ : initialize with $\theta_{0\rightarrow1}$; trained w/ the prompt split $\mathcal{X}_2$ by self-generated responses from the model $\theta_{0\rightarrow1}$.

For evolving prompts (*e.g.*, evolving $\mathcal{X}_1$ to $\mathcal{X}_{\tilde{1}}$), with the calculated informativeness metric for each prompt, we normalize them as the weight to do weighted sampling for a 25% informative subset to get $\mathcal{X}_1^{\texttt{info}}$. We then iterate over in $\mathcal{X}_1^{\texttt{info}}$ and call `EvolInstrut` (Xu et al., 2023) as the plug-in evolving method (with the number of evolutions as 4) using the default mutation templates for (i) in-depth evolving (constraints, deepening, concretizing, increased reasoning steps) and (ii) in-breadth evolving (extrapolation) as implemented in `tasks/evol_instruct/utils.py` of `distilabel==1.3.2`. Next we uniformly select 80% prompts from this evolved dataset and 20% from the original dataset (*i.e.*, the buffer) to form $\mathcal{X}_{\tilde{1}}$. We do not seek extensive parameter search (*e.g.*, the number of evolutions, the evolving ratio) in this stage and encourage future works on exploring this and other plug-in evolving methods. For solver we generate 6 responses per prompt. We use 42 as the random seed.

**Training setting in online `eva` in Table 1.** Plainly put, online `eva` evolves per mini-batch. In Table 1, we have it `eva` works in three phases with the generative buffer. Below we present an easy-to-understand illustration:

1. **Warm-up phase**: Default training with data from $\mathcal{D}$ until the buffer is full. To create the buffer, after each step (training on 8 samples in a mini-batch), we select the top 50% (which is 4) most informative prompts in the mini batch, then evolve $n_{\text{new}}$ versions for each one, and we add only the evolved prompts to the buffer. (Here, we set $n_{\text{new}} = 4$, thus the buffer increases by 16 at each step, until we reach the preset buffer size – which is 3200 currently, *i.e.*, 400 iterations).

2. **Mix-up phase**: From now on, we do a *balanced sampling* from the buffer and $\mathcal{D}$ to form a mini batch for training, where $\text{ratio}_{\text{evol}} = 50\%$: at each optimization step, we sample half of the mini batch from the buffer by weighted

sampling via the informativeness score, and take the rest half i.i.d. from $\mathcal{D}$. We similarly evolve $n_{\text{new}}$ new prompts for the top 50% informative prompts in this batch. We then add the newly generated prompts, and pop the trained ones and the oldest ones from the buffer, to keep a fixed buffer size. Then balanced sampling approach resembles the scheme used in (Ross & Bagnell, 2012; Ball et al., 2023).

3. **Bootstrap phase**: After we run out of samples from $\mathcal{D}$, we will only sample from the buffer, with the same evolving procedure, and add evolved prompts to the buffer and pop out those trained. We take the top 50% from each mini batch, evolve for $n_{\text{new}}^{\text{bootstrap}} = 2$ prompts for each, then pop out with the trained.

The hyper-parameter of the subset set sizes as power of 2 is due to the hardware optimization constraints.

## B. Additional Ablation Experiments for Online `eva`

In the following ablation studies, by keeping other hyper-parameters the same as the above, we show that increasing $n_{\text{new}}$ improves the performance, while a balanced sampling with $\text{ratio}_{\text{evol}} = 50\%$ is generally the most robust.

Table 8: The default setting for the first column of each table is $n_{\text{new}} = 4$ and $\text{ratio}_{\text{evol}} = 50\%$. (Left) **Varying the number of evolved prompts $n_{\text{new}}$.** As we increase $n_{\text{new}}$, we observe a monotonic gain, probably due to the fact that evolving more prompts helps improving the coverage and diversity of training prompts. (Right) **Varying the sampling ratio of evolved prompts in each mini-batch.** In general, we find the balanced sampling strategy is more robust as the training goes on.

| Benchmark ($\rightarrow$) | Arena Hard (**WR** (%)) | |
| --- | --- | --- |
| Setting ($\rightarrow$) | $n_{\text{new}} = 4$ | $n_{\text{new}} = 8$ |
| $\theta_0$: Base Model | 41.3 | 41.3 |
| $\theta_{0 \rightarrow 1}$: RLOO (1x) | 52.6 | 52.6 |
| $\theta_{0 \rightarrow \tilde{1}}$: RLOO-**eva** (1x) | 57.3 | **57.6** |
| $\theta_{0 \rightarrow \tilde{1}}$: RLOO-**eva** (2x) | 60.5 | **61.2** |
| $\theta_{0 \rightarrow \tilde{1}}$: RLOO-**eva** (3x) | 62.4 | **63.0** |
| $\theta_{0 \rightarrow 1}$: RLOO (6x) | 62.7 | 62.7 |

| Benchmark ($\rightarrow$) | Arena Hard (**WR** (%)) | | |
| --- | --- | --- | --- |
| Setting ($\rightarrow$) | $\text{ratio}_{\text{evol}} = 50\%$ | $\text{ratio}_{\text{evol}} = 75\%$ | $\text{ratio}_{\text{evol}} = 25\%$ |
| $\theta_0$: Base Model | 41.3 | 41.3 | 41.3 |
| $\theta_{0 \rightarrow 1}$: RLOO (1x) | 52.6 | 52.6 | 52.6 |
| $\theta_{0 \rightarrow \tilde{1}}$: RLOO-**eva** (1x) | 57.3 | 57.0 | **57.5** |
| $\theta_{0 \rightarrow \tilde{1}}$: RLOO-**eva** (2x) | **60.5** | 59.9 | 59.2 |
| $\theta_{0 \rightarrow \tilde{1}}$: RLOO-**eva** (3x) | **62.4** | 62.0 | 61.3 |
| $\theta_{0 \rightarrow 1}$: RLOO (6x) | 62.7 | 62.7 | 62.7 |

## C. Additional References

In addition to § 5, prior works (Jiang et al., 2024; Xie et al., 2023) have explored signals for adaptive data sampling, which can be integrated into **eva**'s generative pipeline. For the online solver, an alternative to RLOO (Ahmadian et al., 2024) is ReMax (Li et al., 2023). An important future work for **eva** is to have efficient online reinforcement learning of reward models on top of the creator-solver game, on which Russo & Van Roy (2014); Dwaracherla et al. (2024) shed light.

## D. Visualization on Prompt Selection Metric

In **eva**, we assign each prompt an informativeness value, which the creator will use as the weight to sample from the seed prompts for prompt synthesis. In § 4.2.1, we have shown that traditional methods like reward mean and reward variance are less effective as our advantage-based informativeness proxy. The intuition is simple: advantage/regret-based proxy aligns better with the preference optimization objective. We further illustrate that they are statistically different from other choices:

• Figure 8: The distribution of informativeness values shows that reward variance is heavily concentrated at lower values, reward mean is more uniformly scattered, and reward advantage achieves a better balance, providing a broader yet also focused sampling range.

• Figure 9: The *weak correlation* between reward variance and reward advantage shows that variance *cannot* serve as a substitute for advantage as a proxy for informativeness.

We have discussed the contrastive curriculum hypothesis in § H.3 to support using reward advantage in the sense that the induced samples tend to decrease the loss the most in the contrastive optimization. Furthermore, assuming the optimization algorithm can converge to the *more optimal* responses, neither reward mean nor variance directly capture the learning potential of such responses – one may easily construct cases with identical variance yet differ much in reward range –

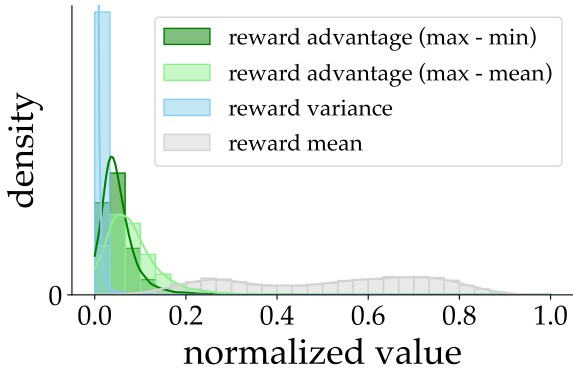
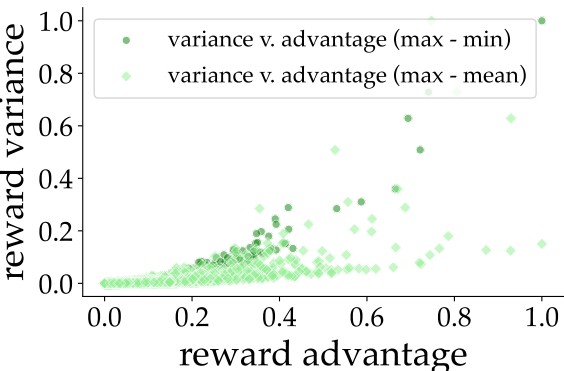

Figure 8: The probability density distributions of informativeness metrics compared in Table 4 – they show different patterns.

Figure 9: The correlation plot for reward advantage (ours) and reward variance – they are only *weakly* correlated.

thus variance fails to distinguish such scenarios. By contrast, reward advantage estimate inherently captures the relative improvement towards better response, and is sensitive to differences in reward range; variants of advantage estimate are commonly used in literature.

## E. Complexity and Quality of Prompts Over Iterations

As in Table 7, there is a gradual improvement of prompt complexity and quality over iterations with **eva**. We sample 10K prompts per iteration, and use the below prompts modified from (Liu et al., 2023b) for the complexity and quality evaluation, with `gemini-1.5-flash` as the scorer:

```
Rank the following questions according to their quality. Your evaluation should consider the following factors: Helpfulness,
↪  Relevance, Accuracy, Depth, Creativity, and Level of detail.
Score each response from 1 to 5: 1: Poor quality, 2: Below average, 3: Average, 4: Good, 5: Excellent.
```

```
Ranking the following questions according to their difficulty and complexity. Use a fixed scoring system: 1: Very simple, 2: Simple,
↪  3: Moderate,  4: Difficult, 5: Very difficult
```

## F. Evolving Instructions

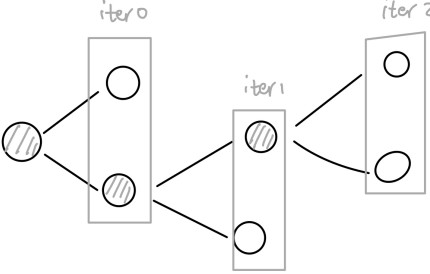

Figure 10: **An illustrative example of tree BoN.** Given a seed prompt, we uniformly sample evolving strategies from below to apply to it, which results in multiple generations; we then proceed with the best one (*e.g.*, by the complexity and quality scorer) as the seed prompt for the next generation. We use this setting in the online **eva** setting as a trial.

```
IN_BREADTH_KEYS = ['persona', 'shift-in', 'shift-out', 'mix', 'abstract']
IN_DEPTH_KEYS = ['constraints', 'deepening', 'concretizing', 'reasoning', 'expansion']
```

```
EVOL_METHODS = {
    # in-breadth evolving
    "persona": (
        "Reframe the #Given Prompt# as if written by a user with a completely different persona, background, or expertise. "
        "Adjust the tone, style, phrasing, or anything you feel proper to reflect this change. "
        "The changes should make the prompt feel like it was authored by someone entirely new."
    ),
    "shift-in": (
        "Shift the high-level idea of the #Given Prompt# to explore a different subdomain or context within the same domain. "
        "Ensure the new topic still challenges the model to reason or provide knowledge relevant to the domain."
    ),
    "shift-out": (
        "Shift the high-level idea of the #Given Prompt# to a completely different topic in a different setting. "
        "The new topic may challenge the model with similar reasoning or contextual understanding but in a novel way."
    ),
    "mix": (
        "Combine the high-level concept of the #Given Prompt# with elements from a different domain. "
        "Introduce novel scenarios or contexts to create diversity while maintaining relevance to the original idea."
    ),
    "abstract": (
        "Turn the #Given Prompt# into a more abstract or generalized version, removing specific details while preserving its intent. "
        "Ensure the new prompt encourages broader, principle-driven reasoning."
    ),

    # in-depth evolving
    "constraints": (
        "Add one or more significant constraints or requirements into the '#Given Prompt#'. "
        "The added constraints must meaningfully alter how the model would respond. "
        "For example, specify additional rules, contexts, or limitations that demand creative adjustments to the response."
    ),
    "deepening": (
        "If the #Given Prompt# contains inquiries about certain issues, increase the depth and breadth of the inquiry. "
        "Make the question require a more detailed, multi-layered, or comprehensive response. "
        "For instance, break the problem into sub-problems or require connections between unrelated concepts."
    ),
    "concretizing": (
        "Replace general concepts in the #Given Prompt# with more specific and detailed concepts. "
        "Ensure that the change makes the problem more defined and concrete, leaving less room for ambiguity. "
        "For example, replace 'a device' with 'a wearable fitness tracker with GPS'."
    ),
    "reasoning": (
        "Add one or more reasoning steps into the '#Given Prompt#'. "
        "Explicitly rewrite it to demand multi-step reasoning or justify intermediate steps in the solution. "
        "For instance, if the original prompt is a simple query, make the response require a step-by-step breakdown of logic or
        ↪  calculations."
    ),
    "expansion": (
        "Expand the #Given Prompt# by including additional perspectives, domains, or layers of complexity. "
        "For example, if the original prompt focuses on a single scenario, add related scenarios or ask the model to compare different
        ↪  situations."
    )
}

INST_IN_DEPTH = (
    "Please act as an expert Prompt Rewriter.\n"
    "Your objective is to rewrite a given prompt into a more complex version "
    "to make those large language models (e.g., gemini) a bit harder to handle.\n"
    "But the rewritten prompt must be reasonable and must be understood and responded by humans.\n"
    "Your rewriting cannot omit the non-text parts such as the table and code in #Given Prompt#, if there is any."
    "You should try your best not to make the #Rewritten Prompt# become verbose, "
    "The #Rewritten Prompt# should be roughly the similar length or a little bit more than that of #Given Prompt#.\n"
    "The #Rewritten Prompt# must sound like a real human user's prompt; DON'T make it sound machine-generated."
    "Specifically, you SHOULD complicate the given prompt using the following method: "
    "\n{method}\n"  # to be formatted
    "The rewritten prompt should reflect meaningful changes across its structure, "
    "ensuring the entire sentence feels sufficiently different from the original. "
    "Again, make sure the rewritten prompt presents a more CHALLENGING TASK."
    "Respond with your rewritten prompt directly. "
    "#Given Prompt#:\n{prompt}\n"  # to be formatted
    "#Rewritten Prompt#:\n"
).lstrip()

INST_IN_BREADTH = (
    "Please act as an expert Prompt Creator.\n"
    "Your objective is to generate a brand-new prompt based on the #Given Prompt#. "
    "The purpose of this task is to promote diversity and generality of training prompts for language models, "
    "helping it practice with varied challenges and perspectives.\n"
    "The LENGTH and complexity of the #Created Prompt# should be similar to that of the #Given Prompt#.\n"
    "The #Created Prompt# must be reasonable, interpretable, and solvable by humans.\n"
    "The #Created Prompt# must sound like a real human user's prompt; DON'T make it sound like machine-generated."
    "Follow the method described below to guide your creation:\n"
    "{method}\n"  # to be formatted
    "The created prompt should reflect meaningful changes across its structure, "
    "ensuring the entire sentence feels sufficiently different from the original. "
    "Respond with your created prompt directly.\n"
    "#Given Prompt#:\n{prompt}\n" # to be formatted
    "#Created Prompt#:\n"
).lstrip()
```

## G. Additional Experimental Results

In general, **eva** maintains the downstream performance and is robust on reasoning-heavy tasks, and the scaling with reward models is more prominent on AlpacaEval, possibly due to training sources for such reward models.

| Method ($\downarrow$) / Dataset ($\rightarrow$) | MUSR-TA | TruthfulQA-Gen | WMDP | GSM8K | GSM-Plus | MMLU-Pro |
|---|---|---|---|---|---|---|
| $\theta_0$: SFT | 38.80 | 34.76 | 58.62 | 24.64 | 18.62 | 52.08 |
| $\theta_{0\rightarrow1}$: DPO | 38.40 | 34.76 | 58.45 | 24.56 | 18.50 | 52.63 |
| $\theta_{1\rightarrow\bar{1}}$:  + eva | 38.40 | 34.15 | 58.40 | 24.26 | 17.96 | **53.03** |
| $\theta_{0\rightarrow1}$: SPPO | 40.80 | 34.15 | 58.72 | 24.79 | 18.42 | 52.70 |
| $\theta_{1\rightarrow\bar{1}}$:  + eva | **41.20** | 34.64 | **58.94** | **25.40** | **18.88** | 52.47 |

Table 9: **Performance on Downstream tasks.**

| Benchmark ($\rightarrow$) | | MT-Bench | | Arena-Hard | AlpacaEval 2.0 | |
|---|---|---|---|---|---|---|
| Method ($\downarrow$) / Metric ($\rightarrow$) | avg. score | 1st turn | 2nd turn | WR (%) | LC (%) | WR (%) |
| $\theta_{0\rightarrow1}$: DPO | 8.66 | 9.01 | 8.32 | 51.6 | 55.01 | 51.68 |
| $\theta_{1\rightarrow\bar{1}}$:  + eva-i (ARMO-8B) | **8.90** | 9.04 | 8.75 | 60.1 | 55.35 | 55.53 |
| $\theta_{1\rightarrow\bar{1}}$:  + eva-i (SKYWORKRM-27B) | 8.75 | 9.07 | 8.43 | **60.3** | **56.12** | **56.40** |

Table 10: **Effect of (pointwise) reward models.**

| Benchmark ($\rightarrow$) | | MT-Bench | | Arena-Hard | AlpacaEval 2.0 | |
|---|---|---|---|---|---|---|
| Method ($\downarrow$) / Metric ($\rightarrow$) | avg. score | 1st turn | 2nd turn | WR (%) | LC (%) | WR (%) |
| $\theta_{0\rightarrow1}$: SPPO | 8.62 | 9.03 | 8.21 | 55.7 | 51.58 | 42.17 |
| $\theta_{1\rightarrow\bar{1}}$:  + eva-i (PAIRRM-0.4B) | 8.78 | **9.11** | 8.45 | 58.9 | 51.86 | 43.04 |
| $\theta_{1\rightarrow\bar{1}}$:  + eva-i (PAIRRM-8B) | **8.89** | 9.08 | **8.70** | **60.2** | **52.71** | **44.52** |

Table 11: **Effect of (pairwise) reward models.**

## H. Additional Experimental Results (as Extensions)

### H.1. Experiments on Different `evolve(·)` Methods

As an addition to Table 2, we have experimented with three different evolve(·) methods, including:

- **SelfInstruct** (Wang et al., 2022): Given seed prompts, variations are created based on criteria such as verb diversity and style blending (mixing interrogative and imperative styles). Unlike EvolInstruct (Xu et al., 2023), which generates prompt variations sequentially, this approach generates independently. We follow the one-shot implementation in `self_instruct.py` of distilabel==1.4.1 and modified the instruction on conciseness so that those newly generated prompts have similar lengths compared to the seed prompts.

- **EvolQuality** and **EvolComplexity** (Liu et al., 2023b): The two methods use the same evolutionary approach (*i.e.*, sequentially generating), but with slightly different meta-instructions for prompt generation, where EvolQuality asks to improve the quality (*i.e.*, helpfulness, relevance, etc) of the seed prompt and EvolComplexity asks to improve the complexity (*i.e.*, increased reasoning steps, etc) of the seed prompt. We follow the implementation in `evol_quality/utils.py` and `evol_complexity/utils.py` of distilabel==1.4.1.

| Model Family ($\rightarrow$) | GEMMA-2-9B-IT | |
|---|---|---|
| Benchmark ($\rightarrow$) | Arena-Hard | |
| Method ($\downarrow$) / Metric ($\rightarrow$) | WR (%) | avg. len |
| $\theta_0$: SFT | 41.3 | 544 |
| $\theta_{0\rightarrow1}$: DPO | 51.6 | 651 |
| $\theta_{1\rightarrow\bar{1}}$:  + eva (evolve(·) = EvolInstruct) | 60.1 | 733 |
| $\theta_{1\rightarrow\bar{1}}$:  + eva (evolve(·) = EvolQuality) | 58.7 | 721 |
| $\theta_{1\rightarrow\bar{1}}$:  + eva (evolve(·) = EvolComplexity) | **60.6** | 749 |
| $\theta_{1\rightarrow\bar{1}}$:  + eva (evolve(·) = SelfInstruct) | 57.2 | 725 |

Table 12: Results of using different evolving methods.

**eva is effective under different evolving methods.** As shown in Table 12, our method brings strong performance gain without training with additional human prompts. Among the experimented methods, we find EvolComplexity shows better results.

## H.2. Experiments on Number of Iterations

As an addition to § 4.2.4, we have experimented with the following settings:

- 10K prompts per iteration with 3 iterations.

- 20K prompts per iteration with 3 iterations (*i.e.*, all seed prompts are used).

- 60K prompts per iteration with 2 iterations (*i.e.*, all seed prompts are used).

Due to time constraints, we did not perform an extensive hyper-parameter search; however, we believe the results presented below sufficiently demonstrate the performance gains achieved by **eva**.

| Model Family ($\rightarrow$) | GEMMA-2-9B-IT | |
| --- | --- | --- |
| Benchmark ($\rightarrow$) | Arena-Hard | |
| Method ($\downarrow$) / Metric ($\rightarrow$) | WR (%) | avg. len |
| $\theta_0$: SFT | 41.3 | 544 |
| $\theta_{0\rightarrow1}$: DPO (10k) | 51.6 | 651 |
| $\theta_{1\rightarrow2}$: DPO (10k) | 59.8 | 718 |
| $\theta_{2\rightarrow3}$: DPO (10k) | 61.2 | 802 |
| $\theta_{1\rightarrow\tilde{1}}$: + **eva** (10k) | 60.1 | 733 |
| $\theta_{\tilde{1}\rightarrow\tilde{2}}$: + **eva** (10k) | 62.0 | 787 |
| $\theta_{\tilde{2}\rightarrow\tilde{3}}$: + **eva** (10k) | 62.2 | 774 |

Table 13: Results of using 10k prompts per iteration (DPO + length-penalized NLL loss).

| Model Family ($\rightarrow$) | GEMMA-2-9B-IT | |
| --- | --- | --- |
| Benchmark ($\rightarrow$) | Arena-Hard | |
| Method ($\downarrow$) / Metric ($\rightarrow$) | WR (%) | avg. len |
| $\theta_0$: SFT | 41.3 | 544 |
| $\theta_{0\rightarrow1}$: DPO (20k) | 53.2 | 625 |
| $\theta_{1\rightarrow2}$: DPO (20k) | 47.0 | 601 |
| $\theta_{2\rightarrow3}$: DPO (20k) | 46.8 | 564 |
| $\theta_{1\rightarrow\tilde{1}}$: + **eva** (20k) | 59.5 | 826 |
| $\theta_{\tilde{1}\rightarrow\tilde{2}}$: + **eva** (20k) | 60.0 | 817 |
| $\theta_{\tilde{2}\rightarrow\tilde{3}}$: + **eva** (20k) | 61.4 | 791 |

Table 14: Results of using 20k prompts per iteration (DPO + length-penalized NLL loss).

| Model Family ($\rightarrow$) | GEMMA-2-9B-IT | |
| --- | --- | --- |
| Benchmark ($\rightarrow$) | Arena-Hard | |
| Method ($\downarrow$) / Metric ($\rightarrow$) | WR (%) | avg. len |
| $\theta_0$: SFT | 41.3 | 544 |
| $\theta_{0\rightarrow1}$: DPO (60k) | 58.9 | 717 |
| $\theta_{1\rightarrow\tilde{1}}$: + **eva** (60k) | 59.6 | 725 |
| $\theta_{\tilde{1}\rightarrow\tilde{1}'}$: + **eva** (60k) | 61.9 | 792 |

Table 15: Results of using 60k prompts per iteration (DPO + length-penalized NLL loss).

**eva can bring robust gains with multiple iterations.** As shown in Table 13, 14, and 15 below, our method presents persistent performance gain over iterations, and concretely surpasses the performance by default DPO training with true human prompts.

### H.2.1. BONUS EXPERIMENTS ON `rewriter`($\cdot$) IN THE LOOP

We present the basic idea here for practitioners to build upon. The motivation comes from the hypotheses derived from § H.2: as the prompts gets harder by evolving, there may be greater demands on the solver's capabilities *compared to earlier iterations*. As such, the solver may not be naively treated the same. One may address this by either scaling up response sampling or introducing meta-instructions to explicitly enhance the solver's reasoning.

We design a proof-of-concept experiment *w.r.t* the latter by adding **rewriter** in **eva**'s solver step. Previously, as in Algo. 1 and § 3.3.1, for each prompt $\mathbf{x}$, we generate multiple responses, and choose the best as $\mathbf{y}_+$ and the worst as $\mathbf{y}_-$ for preference optimization. Now, we add one more rewriting step that attempts to enhance $\mathbf{y}_+$ to be $\mathbf{y}'_+$, by applying a rewriting instruction (Liu et al., 2023b) that asks the solver to alter $\mathbf{y}_+$ with imporved helpfulness, relevance, reasoning depths, creativity and details while keeping the similar length. We then train with $(\mathbf{x}, \mathbf{y}'_+, \mathbf{y}_-)$ for preference optimization. Table 16 shows that adding the rewriter yields concrete performance gains over the default training method, while keeping the training budget and only slightly increasing cost for offline data generation.

| Model Family ($\rightarrow$) | GEMMA-2-9B-IT | |
|---|---|---|
| Benchmark ($\rightarrow$) | **Arena-Hard** | |
| Method ($\downarrow$) / Metric ($\rightarrow$) | **WR** (%) | **avg. len** |
| $\theta_0$: SFT | 41.3 | 544 |
| $\theta_{0\rightarrow1}$: DPO (10k) | 51.6 | 651 |
| $\theta_{1\rightarrow\bar{1}}$:   + **eva** (10k) | **60.1** | 733 |
| $\theta_{1\rightarrow\bar{1}}$:   + **eva** with **rewriter** (10k) | **61.9** | 741 |

Table 16: Results of adding **rewriter** in the **solver** step.

## H.3. Understanding the Informativeness Proxy in Different Intuitive Ways

**Learning potential.**   Our metric intuitively identifies the learning potential of a prompt by measuring the gap between the best and worst response to it from the solver. We reason, that prompts eliciting *both* high-reward and low-reward outcomes, reflect *learnable* tasks where the model is capable of improving but has not yet mastered, thereby implying learning potential (*cf.*, (Jiang et al., 2021b)).

**Worst-case guarantees.**   The minimax-regret objective, by design, leads to solvers that perform robustly across the prompt space, thus gives the worst-case guarantee. While exact equilibrium may not be attainable with approximation, our empirical results in § 4.2.1 demonstrate robustness.

**Auto-curricula for the players.**   We visualize the curriculum induced by **eva** in § 4.2.5. With the stochastic policy, the advantage may be heuristically understood as the reward difference between *a base solver* and *a reference solver*. Rather than optimizing separate solvers (Dennis et al., 2020), we sample multiple times from the same policy to create the pair. In this way, the creator is incentivized to produce new prompts that are just out of the comfort zone of solvers (Chaiklin et al., 2003):

- For overly challenging prompts, both solutions perform poorly, leading to a low proxy.

- For overly easy prompts, the base solution already performs well, again giving a low proxy.

- The optimal strategy is to find prompts that are just beyond the solver's current capability.

**Auto-curricula inherent to Contrastive Optimization.**   Contrastive preference optimization generalizes DPO and a family of algorithms (*c.f.*, (Hejna et al., 2023; Rafailov et al., 2023; Tang et al., 2024)), many of whose losses monotonically decrease as the contrastive ratio increases. To be specific, the DPO (Rafailov et al., 2023) objective for RLHF is:

$$\mathcal{L}_\beta^{\text{DPO}}(\pi_{\boldsymbol{\theta}}) = \sum_{(\mathbf{y}_+, \mathbf{y}_-, \mathbf{x}) \in \mathcal{D}} -\log\left[\sigma\left(\beta \cdot \Delta_{\boldsymbol{\theta}; \text{ref}}^{\mathbf{x}}\right)\right], \tag{7}$$

where we use $+, -$ to denote chosen and rejected responses, and denote the **contrastive ratio** as:

$$\Delta_{\boldsymbol{\theta}; \text{ref}}^{\mathbf{x}} := \log \frac{\pi_{\boldsymbol{\theta}}(\mathbf{y}_+ \mid \mathbf{x})}{\pi_{\text{ref}}(\mathbf{y}_+ \mid \mathbf{x})} - \log \frac{\pi_{\boldsymbol{\theta}}(\mathbf{y}_- \mid \mathbf{x})}{\pi_{\text{ref}}(\mathbf{y}_- \mid \mathbf{x})}. \tag{8}$$

Here, by Table 3 and Eq. 8, the *contrastive ratio* can be written via the *advantage-based proxy*:

$$A^{\star}_{\min}(\mathbf{x}) = \beta \cdot \Delta^{\mathbf{x}}_{\boldsymbol{\theta}^{\star};\,\mathrm{ref}}. \tag{9}$$

By our proxy, we implicitly incentivize the creator to generate prompts that *bring the most contrastive responses*, which decrease the loss the most. This matches the curriculum learning literature, which prioritizes (in **eva**, *generatively* prioritizes) examples with smaller losses for better convergence and generalization (Bengio et al., 2009). We hence suggest the **Contrastive Curriculum Hypothesis**: In contrastive preference optimization, prioritizing prompts with higher contrastive ratio improves sample efficiency and generalization.

We show initial empirical results on this in § 4.2.1 and § 4.2.4.

# I. Examples on Prompts and Model Generations

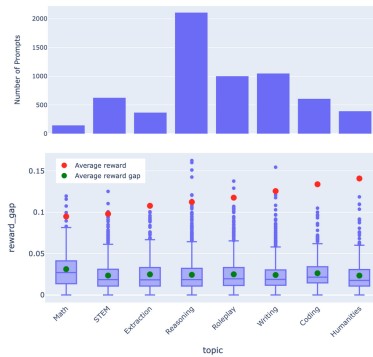

Figure 11: The initial prompt distribution of UltraFeedback by `bart-large-mnli` with 0-shot classification, which is imbalanced. For the reward distribution, the category with lowest average reward has the highest reward gap (*i.e.*, the default informativeness proxy), implying the potential to improve.

Table 17: Baseline examples of evolved prompts from AlpacaEval.

| | |
|---|---|
| initial prompt → | Write me the code for a distributed transaction manager.\nThink step by step and use pseudo code first.\nThen, define interfaces for all involved actors and entities.\nUse Rational Unified approach for this part.\n\nOnly then move on to the actual implementation, class-by-class, and method-by-method.\nMake the code be implemented in C# and follow SOLID principles. |
| evolved #1 → | Craft a suite of syntax for a distributed transaction coordinator. Start with a sequential breakdown in pseudocode format. Following that, establish the protocols for communication and interaction amongst the various participants and components, incorporating the Rational Unified Process methodology.\n\nProceed thereafter to the concrete creation, detailing each class and function. Ensure that the final C# code adheres to the precepts of SOLID and is annotated for clarification and maintainability purposes. |
| evolved #2 → | Devise a comprehensive set of directives and structures for a distributed transaction coordinator architecture. Initiate by formulating a step-by-step algorithmic decomposition in pseudocode. Subsequently, delineate the frameworks for dialog and cooperation between the numerous entities and elements, utilizing the tenets of the Rational Unified Process methodology.\n\nContinue to the actual generation of the code, meticulously detailing every class and corresponding method. Guarantee that the culminating C# code is in strict compliance with SOLID principles and is supplemented with descriptive commentary to enhance future clarity and upkeep, while also validating the code against a set of unit tests to ensure robust functionality. |
| evolved #3 → | Commence by constructing an algorithm in pseudocode that meticulously breaks down the sequential stages for a distributed transaction coordinator system. Progress by outlining the communication protocols and collaborative mechanisms among the myriad components, adhering to the Rational Unified Process approach.\n\nAdvance to crafting the code in C#, ensuring each class and method is elaborated with precision, aligning with SOLID design principles. Integrate thorough comments for heightened maintainability and conduct comprehensive unit testing for assured operational integrity. |
| evolved #4 → | Design a detailed flowchart that represents the intricate steps of an orchestration engine for managing microservices in a cloud-native environment. Begin by detailing the service discovery, load balancing, and fault tolerance strategies, while incorporating the DevOps philosophy.\n\nProceed by developing the implementation in Java, focusing on clean, modular code following object-oriented programming best practices. Accompany your code with clear documentation for easy understanding and perform rigorous integration testing to guarantee robustness. |

| **1st-turn prompt:** | When a number is divided by 10, the remainder is 4. What is the remainder when twice the number is divided by 4? |
|---|---|

**Answer from gemma-2-9b-it-dpo**

Here's how to solve this problem:
1. Understand the first statement:
* "When a number is divided by 10, the remainder is 4" means the number can be represented as 10*k + 4, where k is an integer.
2. Double the number:
* Twice the number is 2*(10*k + 4) = 20*k + 8
3. Divide by 4:
* (20*k + 8) divided by 4 is (20*k)/4 + 8/4 = 5*k + 2
4. Find the remainder:
* Since 5*k is a multiple of 4, the remainder when twice the number is divided by 4 is 2.
Let me know if you'd like to try another example!

**Answer from gemma-2-9b-it-dpo-eva**

Here's how to solve this problem:
1. Understand the first statement:
− "When a number is divided by 10, the remainder is 4" means the number can be represented as 10*k + 4, where k is an integer.
2. Double the number:
− Twice the number is 2*(10*k + 4) = 20*k + 8
3. Divide by 4:
− When (20*k + 8) is divided by 4, the 20*k term is perfectly divisible by 4.
− The remainder will be the same as the remainder when 8 is divided by 4, which is 0.
Therefore, the remainder when twice the number is divided by 4 is 0.

| **2nd-turn prompt:** What about when twice the number is divided by 5? |
|---|

**Answer from gemma-2-9b-it-dpo**

Here's how to solve that:
1. We know: Twice the number is 20*k + 8 (from the previous calculation).
2. Divide by 5: (20*k + 8) divided by 5 is (20*k)/5 + 8/5 = 4*k + 1.6
3. Find the remainder: Since 4*k is a multiple of 5, the remainder when twice the number is divided by 5 is 1.
Let me know if you have any more number theory puzzles!

**Answer from gemma-2-9b-it-dpo-eva**

Here's how to solve that:
1. Twice the number: We know twice the number is 20*k + 8 (from the previous steps).
2. Divide by 5:
* 20*k is divisible by 5.
* 8 divided by 5 leaves a remainder of 3.
Therefore, the remainder when twice the number is divided by 5 is 3.

**Judgment** (by gpt-4)

**Judgment for gemma-2-9b-it-dpo**

The assistant's response is incorrect.
The assistant incorrectly calculated the remainder when twice the number is divided by 5. The assistant calculated the decimal part of the division operation, not the remainder. The correct calculation should be (20n + 8) % 5 = (20n % 5) + (8 % 5) = 0 + 3 = 3. Therefore, the remainder when twice the number is divided by 5 is 3, not 1 as the assistant stated.

The corrected response would be:
1. We know: Twice the number is 20*k + 8 (from the previous calculation).
2. Divide by 5: (20*k + 8) divided by 5 gives us a remainder of (20*k % 5) + (8 % 5) = 0 + 3 = 3.
3. Find the remainder: The remainder when twice the number is divided by 5 is 3.

Rating: [[3]], "score": 3, "turn": 2, "tstamp": 1722737011.7768948

**Judgment for gemma-2-9b-it-dpo-eva**

The assistant's answer is correct and matches the reference answer. The assistant correctly explains that twice the number is 20*k + 8 and that when this is divided by 5, the remainder is 3. The assistant's explanation is clear and easy to understand, making it helpful for the user. There are no mistakes to correct.

Rating: [[10]], "score": 10, "turn": 2, "tstamp": 1722846518.5543818

Table 18: Comparison of model generations on examples from MT-Bench.

---

**1st-turn prompt:**           Imagine you are writing a blog post comparing two popular smartphone models. Develop an outline for the blog post, including key points and subheadings to effectively compare and contrast the features, performance, and user experience of the two models. Please answer in fewer than 200 words.

---

**Answer from gemma-2-9b-it-dpo**

## Smartphone Showdown: [Model A] vs. [Model B]\n\n**I. Introduction**\n* Briefly introduce the two smartphone models being compared, highlighting their popularity and target audience.\n* State the purpose of the comparison: to help readers decide which phone best suits their needs.\n\n**II. Design and Display**\n* Physical dimensions, weight, and build quality comparison.\n* Screen size, resolution, and display technology (e.g., OLED, LCD).\n* Design aesthetics and user preferences.\n\n**III. Performance and Specs**\n* Processor, RAM, and storage comparison.\n* Benchmark scores and real-world performance analysis.\n* Battery life and charging capabilities.\n\n**IV. Camera Capabilities**\n* Rear and front camera specs (megapixels, aperture, features).\n* Image quality comparison in various lighting conditions.\n* Video recording capabilities.\n\n**V. Software and User Experience**\n* Operating system and user interface comparison.\n* App ecosystem and availability.\n* Software features and customization options.\n\n**VI. Price and Value**\n* Compare the price points of both models.\n* Analyze the value proposition based on features and performance.\n\n**VII. Conclusion**\n* Summarize the key differences and similarities.\n* Recommend the best phone for specific user profiles (e.g., budget-conscious, photography enthusiasts, gamers).

**Answer from gemma-2-9b-it-dpo-eva**

## iPhone 14 Pro vs. Samsung Galaxy S23: Which Flagship Reigns Supreme?\n\n**I. Introduction**\n* Briefly introduce both smartphones and their position in the market.\n* State the purpose of the comparison: to help readers decide which phone best suits their needs.\n\n**II. Design and Display**\n* Physical dimensions, weight, and build quality comparison.\n* Screen size, resolution, refresh rate, and display technology (OLED vs. Super AMOLED).\n* Design aesthetics and unique features (e.g., notch vs. punch-hole, color options).\n\n**III. Performance and Camera**\n* Processor comparison (Apple A16 Bionic vs. Snapdragon 8 Gen 2).\n* RAM and storage options.\n* Camera specs: megapixels, lens types, aperture, and special features (e.g., telephoto, macro).\n* Image quality comparison based on sample photos and user reviews.\n\n**IV. Software and User Experience**\n* Operating systems: iOS 16 vs. Android 13.\n* User interface comparison: intuitiveness, customization options, and app ecosystems.\n* Special features: Face ID vs. in-display fingerprint sensor, voice assistants.\n\n**V. Battery Life and Charging**\n* Battery capacity and real-world usage comparison.\n* Fast charging speeds and wireless charging capabilities.\n\n**VI. Price and Value**\n* Compare starting prices and different storage configurations.\n* Assess the overall value for money based on features and performance.\n\n**VII. Conclusion**\n* Summarize the key differences and strengths of each phone.\n* Recommend the best phone for specific user types (e.g., photography enthusiasts, gamers, budget-conscious consumers).

---

**2nd-turn prompt:** Take your previous response and rephrase it as a limerick.

---

**Answer from gemma-2-9b-it-dpo**

Two phones, a showdown so grand,\n[Model A] and [Model B], hand in hand.\nDesign, specs, and price,\nWe'll analyze nice,\nTo help you choose, understand?\n\nLet me know if you'd like me to try filling in the bracketed parts with specific model names!

**Answer from gemma-2-9b-it-dpo-eva**

\nTwo phones, a Pro and a Galaxy,\nFor flagships, they truly are spacy.\nWith cameras and screens,\nAnd processors keen,\nWhich one wins? Read on, and be savvy!\n\nLet me know if you'd like another limerick!

---

**Judgment** (by gpt-4)

---

**Judgment for gemma-2-9b-it-dpo**

The assistant's response is creative and relevant to the user's request. The user asked the assistant to rephrase the previous response, which was a detailed outline for a blog post, into a limerick. The assistant successfully did so, maintaining the essence of the original response in a limerick format. The assistant also offered to further customize the limerick by filling in the bracketed parts with specific model names, showing a willingness to provide additional detail if needed. However, the limerick does not fully capture the depth and detail of the original response, which is understandable given the constraints of the limerick format.

Rating: [[8]], "score": 8, "turn": 2, "tstamp": 1722737024.0327556

**Judgment for gemma-2-9b-it-dpo-eva**

The assistant's response is creative and relevant to the user's request. The assistant successfully rephrased the previous response, which was a detailed outline for a blog post comparing two smartphone models, into a limerick. The limerick is playful and engaging, and it maintains the essence of the original response, which is to compare the two phones and help the reader make an informed decision. The assistant also offers to create another limerick if the user wants, showing a willingness to continue the interaction.

Rating: [[9]], "score": 9, "turn": 2, "tstamp": 1722846542.9437995

---

Table 19: (Continued) comparison of model generations on examples from MT-Bench.

**1st-turn prompt**: Each problem consists of three statements. Based on the first two statements, the third statement may be true, false, or uncertain.\n1. Oranges cost more than apples.\n2. Oranges cost less than bananas.\n3. Bananas cost more than apples and bananas cost more than orange.\nIf the first two statements are true, then the third statement is?

| Answer from gemma-2-9b-it-dpo | Answer from gemma-2-9b-it-dpo-eva |
|---|---|
| Here's how to break down the logic:\n\n* **Statement 1:** Oranges > Apples (Oranges cost more than apples)\n* **Statement 2:** Oranges < Bananas (Oranges cost less than bananas)\n\nThis means the price order is: Bananas > Oranges > Apples\n* **Statement 3:** Bananas > Apples AND Bananas > Oranges\n\nStatement 3 is **partially true**.\n\n* The part "Bananas > Apples" is true based on our price order.\n* The part "Bananas > Oranges" is also true based on our price order.\n\nLet me know if you'd like to try another one! | Here's how to break down the logic:\n\n* **Statement 1:** Oranges > Apples (Oranges cost more than apples)\n* **Statement 2:** Oranges < Bananas (Oranges cost less than bananas)\n\nThese statements tell us that bananas are the most expensive, then oranges, and apples are the least expensive.\n\n* **Statement 3:** Bananas > Apples AND Bananas > Oranges\n\nSince bananas are indeed more expensive than both apples and oranges, statement 3 is **true**.\n\nLet me know if you'd like to try another one! |

**2nd-turn prompt**: If the third statement is true. Is the first statement true, false, or uncertain? Please explain.

| Answer from gemma-2-9b-it-dpo | Answer from gemma-2-9b-it-dpo-eva |
|---|---|
| If the third statement is true, the first statement is **false**.\n\nHere's why:\n\n* **Statement 3:** States that bananas cost more than both apples AND oranges.\n\nThis means the price order is: Bananas > Oranges > Apples.\n* **Statement 1:** States that oranges cost more than apples.\n\nIf bananas are the most expensive, and oranges are second, then oranges cannot be more expensive than apples.\n\nLet me know if you have any other logic puzzles! | If the third statement is true, then the first statement is **uncertain**.\n\nHere's why:\n\n* **Statement 3:** Bananas > Apples AND Bananas > Oranges\n\nThis tells us bananas are the most expensive, but it doesn't explicitly state the relative prices of apples and oranges.\n\n* **Statement 1:** Oranges > Apples\n\nThis statement claims oranges are more expensive than apples.\n\nStatement 3 doesn't contradict statement 1, but it also doesn't confirm it. It's possible that oranges *are* more expensive than apples, but statement 3 alone doesn't give us enough information to be sure. |

**Judgment** (by gpt-4)

| Judgment for gemma-2-9b-it-dpo | Judgment for gemma-2-9b-it-dpo-eva |
|---|---|
| The assistant's response is incorrect. The assistant states that if the third statement is true, the first statement is false. However, the third statement only establishes that bananas cost more than both apples and oranges. It does not provide information about the relative costs of apples and oranges. Therefore, the first statement could be true (if oranges cost more than apples), false (if oranges cost less than apples), or uncertain (if oranges and apples cost the same). The assistant's explanation is flawed because it incorrectly assumes that the order of costs must be Bananas > Oranges > Apples, when in fact the order could also be Bananas > Apples = Oranges or Bananas > Apples > Oranges. \n\nCorrected Assistant's Answer: If the third statement is true, then bananas cost more than apples and bananas cost more than oranges. This means that oranges could still cost more than apples, making the first statement true. However, it's also possible that oranges and apples have the same cost, which would make the first statement false. Therefore, based on the third statement alone, the first statement is uncertain.\n\nRating: [[2]], "score": 2, "turn": 2, "tstamp": 1722737017.2808955 | The assistant's response is correct and helpful. The assistant explains that the first statement is uncertain if the third statement is true, which aligns with the reference answer. The assistant provides a clear explanation of why this is the case, stating that the third statement doesn't contradict the first statement, but it also doesn't confirm it. Therefore, the assistant's response is accurate and informative.\n\nThe assistant's response: "If the third statement is true, then the first statement is **uncertain**. Here's why: Statement 3: Bananas > Apples AND Bananas > Oranges. This tells us bananas are the most expensive, but it doesn't explicitly state the relative prices of apples and oranges. Statement 1: Oranges > Apples. This statement claims oranges are more expensive than apples. Statement 3 doesn't contradict statement 1, but it also doesn't confirm it. It's possible that oranges *are* more expensive than apples, but statement 3 alone doesn't give us enough information to be sure.\n\nRating: [[10]], "score": 10, "turn": 2, "tstamp": 1722846527.2339694 |

Table 20: (Continued) comparison of model generations on examples from MT-Bench.

**1st-turn prompt:**    Implement a function to find the median of two sorted arrays of different sizes with O(1) space complexity and O(n) time complexity.

**Answer from gemma-2-9b-it-dpo**

python\n\def find_median_sorted_arrays(nums1, nums2):\n  """\n Finds the median of two sorted arrays of different sizes.\n\n Args:\n  nums1: The first sorted array.\n  nums2: The second sorted array.\n\n Returns:\n  The median of the two sorted arrays.\n  """\n\n  m, n = len(nums1), len(nums2)\n  if m > n:\n    nums1, nums2, m, n = nums2, nums1, n, m # Ensure m <= n\n\n  imin, imax, half_len = 0, m, (m + n + 1) // 2\n\n  while imin <= imax:\n    i = (imin + imax) // 2\n    j = half_len - i\n    if i < m and nums2[j - 1] > nums1[i]:\n      imin = i + 1 # i is too small, increase it\n    elif i > 0 and nums1[i - 1] > nums2[j]:\n      imax = i - 1 # i is too big, decrease it\n    else:\n      # i is perfect\n\n      if i == 0:\n        max_left = nums2[j - 1]\n      elif j == 0:\n        max_left = nums1[i - 1]\n      else:\n        max_left = max(nums1[i - 1], nums2[j - 1])\n\n      if (m + n) % 2 == 1:\n        return max_left\n\n      if i == m:\n        min_right = nums2[j]\n      elif j == n:\n        min_right = nums1[i]\n      else:\n        min_right = min(nums1[i], nums2[j])\n\n      return (max_left + min_right) / 2.0\n\n# Example usage\nnums1 = [1, 3]\nnums2 = [2]\nmedian = find_median_sorted_arrays(nums1, nums2)\nprint(f"The median of {nums1} and {nums2} is: {median}")\n\n\n**Explanation:**\n\n1. **Handle Unequal Lengths:**\n  – If `nums1` is longer than `nums2`, swap them to ensure `nums1` is always the shorter array.  This simplifies the logic.\n\n2. **Binary Search:**\n  – The core idea is to use binary search on the shorter array (`nums1`).\n  – `imin` and `imax` define the search range for the partition point `i` in `nums1`.\n  – `half_len` calculates the desired total number of elements on the left side of the partition (to find the median).\n\n3. **Partitioning:**\n  – In each iteration, `i` divides `nums1` into two parts:  `nums1[0:i]` and `nums1[i:m]`.\n  – `j` is calculated to divide `nums2` so that the total number of elements on the left side is `half_len`.\n\n4. **Adjusting the Partition:**\n  – We compare elements at the boundaries of the partitions to see if `i` is in the correct position:\n  – If `nums2[j - 1] > nums1[i]`, `i` is too small, so we move `imin` to `i + 1`.\n  – If `nums1[i - 1] > nums2[j]`, `i` is too large, so we move `imax` to `i - 1`.\n\n5. **Finding the Median:**\n  – Once `i` is in the correct position, we find:\n  – `max_left`: The maximum element in the left partitions of both arrays.\n  – `min_right`: The minimum element in the right partitions of both arrays.\n  – If the total length is odd, `max_left` is the median.\n  – If the total length is even, the median is the average of `max_left` and `min_right`.\n\n\n**Time Complexity:** O(log(min(m, n))) because we perform binary search on the shorter array.\n\n\n**Space Complexity:** O(1) as we only use a constant amount of extra space for variables.

**Answer from gemma-2-9b-it-dpo-eva**

python\n\def find_median_sorted_arrays(nums1, nums2):\n  """\n Finds the median of two sorted arrays of different sizes.\n\n Args:\n  nums1: The first sorted array.\n  nums2: The second sorted array.\n\n Returns:\n  The median of the two sorted arrays.\n  """\n\n  m, n = len(nums1), len(nums2)\n  if m > n:\n    nums1, nums2, m, n = nums2, nums1, n, m # Ensure m <= n\n\n  imin, imax, half_len = 0, m, (m + n + 1) // 2\n\n  while imin <= imax:\n    i = (imin + imax) // 2\n    j = half_len - i\n    if i < m and nums2[j - 1] > nums1[i]:\n      imin = i + 1 # i is too small, increase it\n    elif i > 0 and nums1[i - 1] > nums2[j]:\n      imax = i - 1 # i is too big, decrease it\n    else:\n      # i is perfect\n\n      if i == 0:\n        max_left = nums2[j - 1]\n      elif j == 0:\n        max_left = nums1[i - 1]\n      else:\n        max_left = max(nums1[i - 1], nums2[j - 1])\n\n      if (m + n) % 2 == 1:\n        return max_left\n\n      if i == m:\n        min_right = nums2[j]\n      elif j == n:\n        min_right = nums1[i]\n      else:\n        min_right = min(nums1[i], nums2[j])\n\n      return (max_left + min_right) / 2.0\n\n# Example usage\nnums1 = [1, 3]\nnums2 = [2]\nmedian = find_median_sorted_arrays(nums1, nums2)\nprint(f"The median of {nums1} and {nums2} is: {median}")\n\n\n**Explanation:**\n\n1. **Handle Unequal Lengths:**\n  – If `nums1` is longer than `nums2`, swap them to ensure `nums1` is always the shorter array.  This simplifies the logic.\n\n2. **Binary Search:**\n  – The core idea is to use binary search on the shorter array (`nums1`).\n  – `imin` and `imax` define the search range for the partition point `i` in `nums1`.\n  – `half_len` calculates the desired total number of elements on the left side of the partition (to find the median).\n\n3. **Partitioning:**\n  – In each iteration, `i` divides `nums1`, and `j` divides `nums2` such that the total number of elements on the left side is `half_len`.\n\n4. **Adjusting the Search Range:**\n  – If `nums2[j - 1] > nums1[i]`, it means `i` is too small, so we need to move the partition point in `nums1` to the right (`imin = i + 1`).\n  – If `nums1[i - 1] > nums2[j]`, `i` is too large, so we move it to the left (`imax = i - 1`).\n\n5. **Finding the Median:**\n  – When the correct partition is found (`i` is in the right position), we find:\n  – `max_left`: The maximum element on the left side of the partition.\n  – `min_right`: The minimum element on the right side of the partition.\n  – If the total number of elements is odd, `max_left` is the median.\n  – If even, the median is the average of `max_left` and `min_right`.\n\n\n**Time Complexity:** O(log(min(m, n))) because we are performing binary search on the shorter array.\n\n\n**Space Complexity:** O(1) as we are only using a constant amount of extra space for variables.

**Judgment** (by gpt-4)

**Judgment for gemma-2-9b-it-dpo**

The assistant's answer is correct and provides a detailed explanation of the code.  The assistant correctly states that the time complexity is O(log(min(m, n))) and the space complexity is O(1), which aligns with the reference answer.  The assistant also provides an example usage of the function, which is helpful for understanding how to use it.  The assistant's code is a different implementation than the reference answer, but it achieves the same result and meets the same complexity requirements.
However, the assistant's answer does not address the user's original question about achieving O(n) time complexity.  The assistant should have clarified that it's not possible to find the median of two sorted arrays of different sizes with O(1) space complexity and O(n) time complexity, as stated in the answer.

Rating: [[8]], "score": 8, "turn": 1, "tstamp": 1722737514.2011044

**Judgment for gemma-2-9b-it-dpo-eva**

The assistant's answer is correct and provides a detailed explanation of the code.  The assistant correctly states that the time complexity is O(log(min(m, n))) and the space complexity is O(1), which aligns with the reference answer.  The assistant also provides a step-by-step explanation of the code, which is very helpful for understanding the logic behind the solution.  The assistant's code is also correct and will correctly find the median of two sorted arrays of different sizes.  Therefore, there are no corrections needed for the assistant's response.

Rating: [[10]], "score": 10, "turn": 1, "tstamp": 1722846533.772981

Table 21: (Continued) comparison of model generations on examples from MT-Bench.

---

**1st-turn prompt:**       You are given two sorted lists of size m and n.  Implement a function to find the kth smallest element in the union of the two lists with linear complexity.

---

**Answer from gemma-2-9b-it-dpo**

```python
def find_kth_smallest(list1, list2, k):
```
"""\n  Finds the kth smallest element in the union of two sorted lists.\n\n  Args:\n    list1: The first sorted list.\n    list2: The second sorted list.\n    k:  The desired kth smallest element.\n\n  Returns:\n    The kth smallest element in the union of the two lists.\n  """\n\n  m, n = len(list1), len(list2)\n  i, j = 0, 0\n  min_heap = [] \n\n  # Build a min-heap with the first elements of both lists\n  for x in range(min(m, n)):\n    heapq.heappush(min_heap, (list1[x], x, 0)) # (value, list_index, position_in_list)\n    heapq.heappush(min_heap, (list2[x], x, 1))\n\n  # Extract k smallest elements from the heap\n  for _ in range(k - 1):\n    _, _, _ = heapq.heappop(min_heap)\n\n  # The top element of the heap is the kth smallest\n  return heapq.heappop(min_heap)[0]\n\n# Example usage:\nlist1 = [2, 5, 8, 12]\nlist2 = [1, 3, 7, 9, 10]\nk = 5\nkth_smallest = find_kth_smallest(list1, list2, k)\nprint(f"The {k}th smallest element is: {kth_smallest}")\n\n\n**Explanation:**\n\n1. **Initialization:**\n   - 'm' and 'n' store the lengths of the input lists.\n   - 'i' and 'j' are pointers to track the current position in each list.\n   - 'min_heap' is a min-heap used to efficiently store and retrieve the smallest elements encountered so far.\n\n2. **Building the Min-Heap:**\n   - The code iterates through the first 'min(m, n)' elements of both lists.\n   - For each element, a tuple '(value, list_index, position_in_list)' is pushed onto the 'min_heap'.  This tuple helps us keep track of the element's value, the list it belongs to, and its original position.\n\n3. **Extracting k-1 Smallest Elements:**\n   - The 'for' loop runs 'k - 1' times, effectively extracting the 'k-1' smallest elements from the 'min_heap'.\n\n4. **Finding the kth Smallest:**\n   - After the loop, the top element of the 'min_heap' contains the 'k'th smallest element.\n   - 'heapq.heappop(min_heap)' removes and returns this element.\n\n**Time Complexity:**\n- Building the min-heap:  O(m + n)\n- Extracting k-1 elements:  O(k * log(k))\n- Overall:  **O(m + n + k * log(k))**, which is approximately **O(m + n)** since 'k' is usually much smaller than 'm + n'.\n\n\nLet me know if you have any other questions!

**Answer from gemma-2-9b-it-dpo-eva**

```python
def find_kth_smallest(list1, list2, k):
```
"""\n Finds the kth smallest element in the union of two sorted lists.\n\n Args:\n   list1:  The first sorted list.\n   list2:  The second sorted list.\n   k:  The desired rank of the smallest element.\n\n Returns:\n   The kth smallest element in the union of the lists.\n """\n\n i = 0 # Index for list1\n j = 0 # Index for list2\n merged = [] # Store the merged elements\n\n while i < len(list1) and j < len(list2):\n   if list1[i] <= list2[j]:\n   merged.append(list1[i])\n   i += 1\n   else:\n   merged.append(list2[j])\n   j += 1\n\n # Add any remaining elements from list1\n while i < len(list1):\n merged.append(list1[i])\n   i += 1\n\n # Add any remaining elements from list2\n while j < len(list2):\n merged.append(list2[j])\n   j += 1\n\n return merged[k - 1] # Return the kth element (index k-1)\n\n# Example usage:\nlist1 = [2, 5, 8, 12]\nlist2 = [1, 3, 7, 9, 10]\nk = 5\nkth_smallest = find_kth_smallest(list1, list2, k)\nprint(f"The {k}th smallest element is: {kth_smallest}")\n\n\n**Explanation:**\n\n1. **Initialization:**\n   - 'i' and 'j' are index pointers for 'list1' and 'list2' respectively, starting at 0.\n   - 'merged' is an empty list to store the merged sorted elements.\n\n2. **Merging:**\n   - The 'while' loop continues as long as both 'i' and 'j' are within the bounds of their respective lists.\n   - In each iteration, it compares the elements at 'list1[i]' and 'list2[j]'.\n   - The smaller element is appended to 'merged', and the corresponding index ('i' or 'j') is incremented.\n\n3. **Adding Remaining Elements:**\n   - After one list is exhausted, the remaining elements from the other list are directly appended to 'merged'.\n\n4. **Returning the kth Element:**\n   - Finally, 'merged[k - 1]' is returned, which represents the kth smallest element in the union (remember that list indices start from 0).\n\n**Linear Complexity:**\n- The algorithm iterates through each element in both lists at most once during the merging process.\n- The time complexity is therefore O(m + n), which is linear in the total number of elements (m + n), making it efficient for large lists.

---

**Judgment** (by gpt-4)

---

**Judgment for gemma-2-9b-it-dpo**

The assistant's answer is incorrect.  The user asked for a solution with linear complexity, but the assistant provided a solution with a time complexity of O(m + n + k * log(k)), which is not linear.  The assistant's solution uses a min-heap, which is not necessary and adds unnecessary complexity to the solution.

Rating: [[2]], "score": 2, "turn": 1, "tstamp": 1722737030.6411152

**Judgment for gemma-2-9b-it-dpo-eva**

The assistant's answer is correct and helpful.  It provides a Python function that implements the two-pointer approach to find the kth smallest element in the union of two sorted lists.  The assistant's function is slightly different from the reference, but it achieves the same goal.  The assistant's function merges the two lists into a new list and then returns the kth element from the merged list.  The assistant also provides a detailed explanation of how the function works and confirms that the time complexity is linear, which is what the user asked for.  However, the assistant's function uses extra space to store the merged list, which is not necessary and not as efficient as the reference.

Rating: [[8]], "score": 8, "turn": 1, "tstamp": 1722846543.867601

---

Table 22: (Continued) comparison of model generations on examples from MT-Bench.

---

**2nd-turn prompt: Does there exist an algorithm with better time complexity? If so, implement it.**

---

**Answer from gemma-2-9b-it-dpo**

You are absolutely right! There's a more efficient algorithm with a time complexity of **O(m + n)** without using a heap.\n\nHere's the implementation:\n\npython\n\def find_kth_smallest_optimized(list1, list2, k):\n """\n Finds the kth smallest element in the union of two sorted lists\n with O(m + n) time complexity.\n\n Args:\n list1: The first sorted list.\n list2: The second sorted list.\n k: The desired kth smallest element.\n\n Returns:\n The kth smallest element in the union of the two lists.\n """\n m, n = len(list1), len(list2)\n i, j = 0, 0\n result = []\n\n # Merge the lists while keeping track of the kth smallest\n while len(result) < k:\n if i < m and j < n:\n if list1[i] <= list2[j]:\n result.append(list1[i])\n i += 1\n else:\n result.append(list2[j])\n j += 1 elif i < m:\n result.append(list1[i])\n i += 1\n else:\n result.append(list2[j])\n j += 1\n\n return result[k - 1]\n\n# Example usage:\nlist1 = [2, 5, 8, 12]\nlist2 = [1, 3, 7, 9, 10]\nk = 5\nkth_smallest = find_kth_smallest_optimized(list1, list2, k)\nprint(f"The {k}th smallest element is: {kth_smallest}")\n\n\n**Explanation:**\n\n1. **Initialization:**\n - `m` and `n` store the lengths of the lists.\n - `i` and `j` are pointers for `list1` and `list2`, respectively.\n - `result` is a list to store the merged and sorted elements.\n\n2. **Merging and Tracking:**\n - The `while` loop continues until `k` elements are in the `result` list.\n - In each iteration, it compares the current elements from `list1` and `list2`.\n - The smaller element is appended to `result`, and the corresponding pointer (`i` or `j`) is incremented.\n - If one list is exhausted, the remaining elements from the other list are appended to `result`.\n\n3. **Returning the kth Smallest:**\n - Finally, `result[k - 1]` contains the `k`th smallest element.\n\n**Time Complexity:**\n- The algorithm iterates through both lists once, resulting in a time complexity of **O(m + n)**.\n\n\nLet me know if you have any more questions!

**Answer from gemma-2-9b-it-dpo-eva**

You are right to ask! While the previous algorithm has linear time complexity, we can achieve an even better solution with **O(log(min(m, n)))** time complexity using a slightly more sophisticated approach.\n\nThe key idea is to leverage the sorted nature of the lists and use a **binary search** strategy.\n\nHere's the implementation:\n\n##\npython\n\def find_kth_smallest_optimized(list1, list2, k):\n """\n Finds the kth smallest element in the union of two sorted lists with O(log(min(m, n))) time complexity.\n\n Args:\n list1: The first sorted list.\n list2: The second sorted list.\n k: The desired rank of the smallest element.\n\n Returns:\n The kth smallest element in the union of the lists.\n """\n m, n = len(list1), len(list2)\n if m > n:\n list1, list2, m, n = list2, list1, n, m # Ensure list1 is the shorter one\n\n left, right = 0, m\n while left <= right:\n partitionX = (left + right) // 2\n partitionY = k - partitionX\n\n maxLeftX = list1[partitionX - 1] if partitionX > 0 else float('-inf')\n minRightX = list1[partitionX] if partitionX < m else float('inf')\n\n maxLeftY = list2[partitionY - 1] if partitionY > 0 else float('-inf')\n minRightY = list2[partitionY] if partitionY < n else float('inf')\n\n if maxLeftX <= minRightY and maxLeftY <= minRightX:\n if partitionX == 0:\n return list2[partitionY - 1]\n elif partitionY == 0:\n return list1[partitionX - 1]\n else:\n return max(list1[partitionX - 1], list2[partitionY - 1]) elif maxLeftX > minRightY:\n right = partitionX - 1 # Too many elements on the right side of list1\n else:\n left = partitionX + 1 # Too many elements on the right side of list2\n\n\n**Explanation:**\n\n1. **Shorten Lists:** If `list1` is longer than `list2`, swap them to ensure `list1` is always the shorter one. This simplifies the logic.\n\n2. **Binary Search:**\n - `left` and `right` pointers define the search space within `list1`.\n - `partitionX` is the middle point in `list1`, and `partitionY` is calculated to divide the remaining elements in `list2` such that `partitionX + partitionY = k`.\n - `maxLeftX`, `minRightX`, `maxLeftY`, and `minRightY` represent the largest element to the left and smallest element to the right of the partitions in both lists.\n\n3. **Comparison and Adjustment:**\n - If the `maxLeftX` is less than or equal to `minRightY` and `maxLeftY` is less than or equal to `minRightX`, it means the kth smallest element lies within the current partitions. We return the larger of `maxLeftX` and `maxLeftY`.\n - If `maxLeftX` is greater than `minRightY`, it indicates that we need to search in the left subarray of `list1` (i.e., decrease `right`).\n - Otherwise, we search in the right subarray of `list1` (i.e., increase `left`).\n\n4. **Termination:** The binary search continues until `left` crosses `right`, indicating the correct partition has been found.\n\n**Time Complexity:**\n- The binary search divides the search space in half at each step.\n- Therefore, the time complexity is O(log(min(m, n))), which is significantly faster than linear time for large lists where one list is much smaller than the other.

---

**Judgment** (by gpt-4)

---

**Judgment for gemma-2-9b-it-dpo**

The assistant's first response to the user's question is incorrect. The user asked for a function to find the kth smallest element in the union of two sorted lists with linear complexity. The assistant provided a function that uses a min-heap, which has a time complexity of O(m + n + k * log(k)). This is not linear complexity, as the user requested. The assistant's second response to the user's question is also incorrect. The user asked if there exists an algorithm with better time complexity than linear. The assistant provided a function that has a time complexity of O(m + n), which is linear, not better than linear. The assistant should have provided a function that uses a binary search approach, which has a time complexity of O(log(m) + log(n)), as the reference answer correctly does.

Rating: [[2]], "score": 2, "turn": 2, "tstamp": 1722737031.5033472

**Judgment for gemma-2-9b-it-dpo-eva**

The assistant's response is correct and helpful. The assistant provided a Python function that uses a binary search approach to find the kth smallest element in the union of two sorted lists. The assistant also provided a detailed explanation of how the function works, including the time complexity of the function. The assistant's answer matches the reference answer in terms of the algorithm used and the explanation provided. The assistant also correctly identified that the binary search approach has a better time complexity than the linear approach. However, the assistant's code has a minor issue. The assistant's code does not handle the case when `k` is greater than the sum of the lengths of the two lists. In such a case, the code will raise an `IndexError`. This can be fixed by adding a check at the beginning of the function to return `None` if `k` is greater than the sum of the lengths of the two lists.

Rating: [[9]], "score": 9, "turn": 2, "tstamp": 1722846556.6828268

---

Table 23: (Continued) comparison of model generations on examples from MT-Bench.

