# OpenReview forum: "Reward-Guided Prompt Evolving in Reinforcement Learning for LLMs"
_ICML.cc/2025/Conference — ICML 2025 poster_

### Official Review · Reviewer_irNB · 2025-03-09

**Overall Recommendation:** 4

**Summary:**

This paper presents eva, a new minimax algorithm for RLHF which pushes beyond the static prompt set used by the majority of RLHF algorithms. In eva, the creator is trained to generate prompts that are solvable by the the solver. The largely empirical work focuses on evaluating a large number of design choices through extensive experimental evaluation.

**Claims And Evidence:**

The claims made in the paper surrounding eva's performance are largely substantiated by extensive experiments.

However, one component that I believe would be important to understanding eva's claimed performance would be the different prompt/datasets used to train the reward model versus used for RL. I have listed some questions surrounding this in the "questions" section.

**Essential References Not Discussed:**

N/A

**Experimental Designs Or Analyses:**

The work is thoroughly evaluated on a number of different axes. I commend the authors for their work in setting up all of these experiments.

One aspect that was less clear to me were the differences between the prompt datasets used to train the reward model versus prompt datasets used for the RL component. From looking at experiments in Appendix F, it seems like eva's gains versus DPO degrade substantially when DPO is given larger preference datasets.

I think it is most fair to evaluate DPO and eva with access to the same base prompt dataset. If the reward function used by EVA to select prompts is trained on all the data, this seems like an unfair advantage in the evaluation.

**Methods And Evaluation Criteria:**

The presentation of the method is clear and the work provides sufficient justification for the objective presented in Eq. 2 through connections to minimax regret.

The chosen evaluations make sense as they are standard evaluations used for RLHF / Alignment.

**Other Comments Or Suggestions:**

* I can tell the authors were crammed for space, but I think it is nicer when papers write full works e.g. "with" instead of "w/" (Line 198 col 2)

* Line 229 col 2 "and can compete default training". Is there a missing word here?

**Other Strengths And Weaknesses:**

-

**Questions For Authors:**

* Eva appears to require having an explicit reward model. Could the authors detail how that reward model is trained?

* Could the authors provide more details on how they evaluate EVA? What subset of prompts are used from ultrafeedback?

* How does eva compare as the number of prompts in the initial set increase?

* Could the authors do a better job clarifying what data is used for training the reward model used for EVA versus the data used for RL? It seems like these might be different in places. Does this create a problem where the reward function must have more data to be sufficiently generalizable to generated prompts?

**Relation To Broader Scientific Literature:**

Though prior work exists on both prompt evolution and evolution based reward design, eva presents an exciting proof of concept for dataset expansion for post-training RL. This demonstration is at the highest scale I have seen, and I believe would be of value to the community.

**Theoretical Claims:**

The work does not make any theoretical claims.

---

> ### Author Rebuttal · Authors · 2025-04-01
>
> We sincerely thank the reviewer for the detailed review and insightful questions. Below we provide a high-level summary with detailed rebuttal, and will add relevant discussions in the reivisions.
>
> ---
>
> > **Q2 & Q3**: *Could the authors provide more details on how they evaluate `eva`? What subset of prompts are used from ultrafeedback? How does `eva` compare as the number of prompts in the initial set increases?*
>
> **A**:  As discussed in Section 4, we evaluate `eva` on off-the-shelf benchmakrs including [AlpacaEval 2.0](github.com/tatsu-lab/alpaca_eval), [MT-Bench](github.com/lm-sys/FastChat/blob/main/fastchat/llm_judge), and [Arena-Hard](github.com/lmarena/arena-hard-auto). We train `eva` with the train split of UltraFeedback; following standard practices in iterative RLHF, we shuffle the full training set and divide it into equal-sized subsets for each iteration. In Appendix F.2., we have experimented with varying number of prompts in the initial set from 10K to 20K and 60K, and show that `eva` can consistently bring robust gains across multiple iterations.
>
> > **Q1 & Q4**:  (**i**) Could the authors detail how the reward models are trained? (**ii**) What data is used for training the reward model *v.s.* the data used for training the policy model (*i.e.*, the solver)? (**iii**) Does this create a problem where the reward model must have more data to be sufficiently generalizable to generated prompts?
>
> **A**:   (**i**) As discussed in Section 3 and 4, we assume a fixed reward model as the oracle for human preferences during training. We have evaluated our method under different reward models ([ArmoRM-8B](https://arxiv.org/pdf/2406.12845), [SkyworkRM-27b](https://arxiv.org/pdf/2410.18451), [PairRM-0.4B](arxiv.org/abs/2306.02561), and [PairRM-8B](huggingface.co/RLHFlow/pair-preference-model-LLaMA3-8B)). (**ii**) The data used for training the reward model can be different from the data used to train the solver. For example, the prompts used to train ArmoRM-8B and PairRM-8B are selected from UltraFeedback, HelpSteer, OpenOcra, UltraInteract, Capybara and DIBT-10K; the prompts used to train SkyworkRM-27B are 80K prompts selected from  HelpSteer2,  OffsetBias, WildGuard (adversarial), and Magpie. (**iii**) We believe continual training of reward models is an important future work to enhance the robustness of `eva`. As the creator generates more prompts that diverge from the initial training prompt set (in our case, subsets of UltraFeedback), a fixed RM may be struggle to generalize reliably, especially if its training data does not cover the evolving prompt distribution. While our experiments show that `eva` remains effective under multiple fixed RMs, we view continual refinement of reward models (e.g., by online updates or co-training with the evolving policy) as important future works for improving the long-term robustness of `eva`.  Note that reward models may be trained more efficiently and generalize better than policy models, as they only produce scalar scores or rankings, thus it does not necessarily require *more* data (or prompts) than the policy to remain effective.
>
>
> > **Comments:** *It is nicer when papers write full works, e.g., "with" instead of "w/". Line 229 col 2 "and can compete default training". Is there a missing word here?*
>
> **A**: Thanks so much for the suggestions! We will carefully revise each and every abbreviation and make sure the writing is clear. And yes, for Line 229, we meant that `eva` with only 1× raw data throughout training can compete with default training -- which uses 5× more human prompts and no evolved data -- by achieving comparable or better results.

---

> > ### Comment · Reviewer_irNB · 2025-04-02
> >
> > Thanks for answering my questions! I will maintain my score.

---

### Official Review · Reviewer_QZbz · 2025-03-13

**Overall Recommendation:** 4

**Summary:**

This paper studies a new paradigm for post-training where prompts are sampled adaptively. In particular, this paper proposes eva, in which a creator is addtionally introduced to select prompts for the solver to optimize. It provides extensive empirical results to show the advantage of eva.

## update after rebuttal

I have reviewed the authors' updated formulation, which is now more sound than the submitted version. The empirical ablations also address my concerns regarding effectiveness. Below are additional comments on the rigor of the formulation:

- Gap between Problem 1 and the min-max regret formulation: In Problem 1, the objectives of the two players differ, whereas in the min-max Nash game formulation, they are the same.

- Soundness of Problem 1: The formulation appears reasonable, though the lack of a prior definition for $\pi_{true}$ remains a subtle issue. I guess the issue is not from the statistical side that we cannot draw samples from $\pi_{true}$. Instead, the fundamental concern is that there is no clear criterion for defining the of optimality of $\pi_{true}$. This is analogous to many optimization problems where the optimal solution is unknown a priori but is instead implicitly defined by the objective function.

- Min-max regret formulation: Under the assumption that the solver has strong representation and optimization power, the optimal best response would simply select the action with the highest reward, regardless of the creator’s design. To avoid this trivial case, additional assumptions (e.g., limited representation or optimization power) should be introduced.

Overall, I find the empirical algorithm `eva` reasonable and potentially valuable to the community, though the theoretical formulation should be more rigorous. Given these considerations, I have updated my review score to 4, and I hope that the authors could either refine these aspects or remove unproper formulation.

**Claims And Evidence:**

The story is imaginative, but its scientific foundation is questionable.

Issue with Problem 1: The problem highlighted in Problem 1 does not make sense to me. Without proper regularization for the creator, the creator may simply choose the simplest prompt for the solver, leading to an equilibrium that is essentially meaningless. Although the paper addresses this issue in Section 3.1 and introduces actor regularization, it still does not resolve my concerns. Regularization should not be treated as a central element of game design. Therefore, I believe the game is fundamentally flawed in its design. Moreover, if regularization is indeed crucial, the paper should discuss it more thoroughly in the main text. Currently, I see no significant discussion of this in the main text.

Discrepancy between Problem 1 and the Formulation: There is a notable gap between Problem 1 and the formulation presented in Section 2. In Problem 1, the game is collaborative, with both layers aiming to maximize the same objective (i.e., a max-max formulation). However, in Section 2, the paper shifts to a min-max game formulation, where the creator appears to act adversarially. This inconsistency suggests that the formulation does not actually solve the problem proposed in Problem 1.

**Essential References Not Discussed:**

This paper provides a comprehensive review of previous works. However, I would like to highlight several key points and relevant literature that should be discussed.

The core idea of regret maximization for the creator bears strong similarities to the following two works, which deserve discussion [1, 2].

The paper should also consider discussing the work [3], which presents an information-theoretic approach to data collection.

The gradient estimator used for the actor appears to be similar to the ReMax algorithm [4].

To my knowledge, there are two main paradigms for adaptive sampling:
- Information-Seeking (Min-Max Formulation): This approach may demonstrate advantages over uniform sampling in compute-limited settings.
- Transferability-Based Sampling: This approach aims to select prompts from a large source pool based on representative samples from the target domain. For example, see [5]

[1] Jiang, Yiding, et al. "Adaptive data optimization: Dynamic sample selection with scaling laws." arXiv preprint arXiv:2410.11820 (2024).

[2] Mindermann, Sören, et al. "Prioritized training on points that are learnable, worth learning, and not yet learnt." International Conference on Machine Learning. PMLR, 2022.

[3] Dwaracherla, Vikranth, et al. "Efficient exploration for LLMs." arXiv preprint arXiv:2402.00396 (2024).

[4] Li, Ziniu, et al. "ReMax: A simple, effective, and efficient reinforcement learning method for aligning large language models." arXiv preprint arXiv:2310.10505 (2023).

[5] Xie, Sang Michael, et al. "Data selection for language models via importance resampling." Advances in Neural Information Processing Systems 36 (2023): 34201-34227.

**Experimental Designs Or Analyses:**

Yes, I reviewed the experiment details and found that the proposed approach, as described in Appendix A, appears to be rather heuristic. For instance, it uses a specific number of prompts (e.g., 4 and 16) without clear justification. This raises concerns about the generalizability of the approach: is the demonstrated performance highly sensitive to this hyperparameter? Without further analysis or ablation studies, it is difficult to assess whether the results are robust or overly dependent on this particular configuration.

**Methods And Evaluation Criteria:**

Not applicable

**Other Comments Or Suggestions:**

I find that Remark 1 does not provide any new insights beyond reiterating the definition.

**Other Strengths And Weaknesses:**

The results are presented in a well-organized format; however, certain parts lack clear explanation in the main text, making it difficult to fully assess the scientific value of the work. For instance:

- The details of creator regularization in Section 3.1 are not sufficiently elaborated.

- The creator optimization step in Section 3.3.2 is not clearly explained.

Additionally, the paper fails to discuss simple baselines (e.g., uniform sampling) in the formulation and does not provide a thorough analysis of why and under what conditions the proposed approaches are expected to outperform these baselines. Addressing these gaps would significantly strengthen the paper's scientific rigor and practical relevance.

**Questions For Authors:**

From my understanding, the min-max formulation is advantageous in compute-limited settings, as it can quickly identify samples whose loss decreases rapidly. However, from a theoretical standpoint, it is clear that the optimal strategy for the creator is to uniformly select all prompts if the solver is provided with sufficient computational resources to optimize under each prompt. In such cases, uniform sampling is expected to performs well. I note that the experiments are conducted with relatively small sample sizes (e.g., 10k). I am curious about the performance advantage of the proposed approach over uniform sampling when the data size is significantly larger (e.g., 1M, as seen in commercial-level LLM products).

Additionally, I am unclear about the design of the prompt buffer:

- Is it incremental (e.g., 10k → 10k + 10k = 20k → 20k + 10k = 30k) in each iteration?

- Or is it fixed (e.g., 10k, 10k, 10k) in each iteration?

I feel confused about this setting. In the latter case, I suspect there may be a knowledge forgetting issue, as previous prompts are discarded in later stages of training.

Finally, I would like to see ablation studies on the hyperparameter choices for online EVA mentioned in Appendix A. The current details provided are too heuristic and lack justification, making it difficult to assess the robustness and generalizability of the approach.

**Relation To Broader Scientific Literature:**

The problem of adaptive prompt selection studied in this paper is quite interesting; however, its scientific value remains questionable.

**Theoretical Claims:**

Not applicable

---

> ### Author Rebuttal · Authors · 2025-04-01
>
> We sincerely thank the reviewer for the detailed and thoughtful review, which helps us a lot in shaping a better submission.
>
> ---
> **Overview:** We would like to clarify several potential misunderstandings in the review:
> 1. **Problem 1 and Minimax Game:**
>     - (**i**) "*Problem 1 is a max-max collaborative game.*" --> Incorrect. Problem 1 is a joint optimization problem, whose solution may be approximated in either a min-max or max-max way.
>     - (**ii**) "*Creator regularization is not implemented.*" --> Incorrect. Creator regularization is explicitly achieved through regret maximization (as in Section 3.1 and 3.2).
>     - (**iii**) "*The minimax regret formulation does not actually solve the Problem 1.*" --> Inaccurate. The minimax regret formulation provides a worst-case optimal solution to Problem 1 (as in the discussions under Remark 1).
> 2. **Adaptive Prompt Sampling:**
>     - "*It is clear that the optimal strategy is to uniformly select all prompts.*" --> Potentially misleading. The assumption on unlimited resources can be impractical, as efficiency is a key bottleneck in training large models. Even w/ unlimited resources, uniform sampling is only optimal under strong assumptions (e.g., iid data, perfect RM, ...), which is impractical. We also cited a rich literature on online learning and network optimization showing uniform sampling can be sub-optimal and may lead to worse local minima (as in Table 5, 4.2.1 and 4.2.4).
>
> ---
>
> **1. Rebuttal on Problem 1 and the Minimax Game:**
> (**i**) Problem 1 captures the general objective on optimizing the language model for it to perform well with regard to some potentially unknown prompt distribution (we will add $\pi _{\mathsf{true}}(\cdot)$ inside the regularization to emphasize the discussion in 3.1.)
> $$
> \max _{\phi, \boldsymbol{\theta}} \mathcal{J}(\phi, \theta) := \mathbb{E} _{  x \sim \pi _\phi(\cdot)}  [\mathbb{E} _{  y \sim \pi _{\boldsymbol{\theta}}(\cdot \mid   x)}[r(  x,   y)]-\beta_1 \cdot \mathbb{D} _{\mathsf{KL}} [\pi _{\boldsymbol{\theta}}(  y \mid   x) \| \pi _{\mathsf {base}}(  y \mid   x)  ]  ] + \mathcal{R} (\pi _\phi(\cdot), \pi _{\mathsf{true}}(\cdot)  ).
> $$
>
> This is a joint optimization problem, and itself is **not directly a collaborative (nor competitive) game**. It can be intractable and may be approximated differently in max-max or max-min way by alternating optimization (cf., GAN). With $c$ for creator, $s$ for solver, and $c _{t}$ for target, some choices may be:
> - $\max _{s} \max _{c} f _c(s) - \mathsf{KL}(c, c _{t})$
> - $\max _{s} \min _{c} f _c(s) + \mathsf{KL}(c, c _{t})$
>
> We believe the current joint optimization formulation is more general, and can induce different practical algorithms.
>
> (**ii**) / (**iii**) When the target true distribution is unknown, the problem is a classical *decision under ignorance* problem (with partial knowledge, it can become decision under uncertainty and strategies like posterior sampling can be applied) (Jiang, 2023; Peterson, 2017), where we need find a propoer decision rule. Here, we seek the *optimal solution under the worst-case*, and this is why we design a game, where **creator regularization** is explicitly achieved by **regret maximization**.  The game provides an approximation to the worst-case-optimal solution of Problem 1.
>
> It may be subjective to claim "regularization should not be a central element of game" and incorrect to derive "the game is fundamentally flawed". Here, the regularization is applied to the joint optimization problem, not the minimax regret game.
>
> ---
>
> **2. Rebuttal on Online `eva` Settings:**
> Buffer subset sizes are chosen as powers of 2 for hardware efficiency, as we run on a single machine with 8 GPUs.
>
> We will add further results in anonymous.4open.science/r/eva-i.
>
> ---
>
> **3. Rebuttal on Additional References:**
> Thanks for the wonderful suggestions! We will add them in our revised paper. Some discussions:
> - [2] has been cited and discussed in 4.2.1.
> - [3] is on training reward models, which differs from our main theme. We have cited earlier Thompson Sampling work to reflect this area.
> - We do not assume access to target as in [5].
>
> ---
> **4. Rebuttal on Adaptive Sampling:**
> Please see our overview in the beginning.
>
> We use fixed design for the buffer, which is standard in iterative RLHF (e.g., SimPO). In practice, we found "knowledge forgetting" less of an issue (than overfitting), likely due to differences in supervised learning and RL.
>
> ---
>
> **References on Fundamentals:**
> Jiang, M. (2023). Learning Curricula in Open-Ended Worlds.
> Peterson, M. (2017). An introduction to decision theory.
> Orabona, F. (2023). A modern introduction to online learning.
>
> ---
> It is a great pleasure to have the opportunity to learn from a different mind. We believe the rebuttal have sufficiently addressed the concerns. We sincerely hope the reviewer may reconsider the rating on `eva`, and we are happy to discuss further on any potential future works.

---

> > ### Comment · Reviewer_QZbz · 2025-04-04
> >
> > Thanks for the clarification!
> >
> > **Response to Problem 1 and the Minimax Game:**  Thank you for your clarification. However, I disagree with the rebuttal's argument. When you define the problem as a joint optimization problem, it is effectively a single maximization problem—albeit with optimization variables partitioned into two blocks that share the same objective. In contrast, a minimax optimization problem involves two distinct objectives.
> >
> > To illustrate this distinction, consider the function $ f_c(s) = c \cdot s $, where $ c $ and $ s $ are scalars for simplicity.
> >
> > - If we solve $ \max_{c} \max_{s} f_c(s) $, the solution is $ c = s = \infty $, yielding an optimal value of $ \infty $.
> > - However, if we solve $ \max_{c} \min_{s} f_c(s) $, the solution becomes $ c = s = 0 $, with an optimal value of $ 0 $.
> >
> > These two formulations clearly lead to different outcomes, demonstrating that they are not equivalent.
> >
> > **Response to the Regularization Concern:**  I remain unclear about the regularization aspect. Since the main paper does not discuss it in sufficient detail, could you formulize the regularition term in Equation (2)? Also, it would better to explicitly connect the regularization in Equation (2) to the steps in Algorithm 1.
> >
> > **Response to the Empirical Results:**   I was unable to access the repository link provided, as it appears to be expired. Could you share an updated or alternative link?

---

> > > ### Author Response · Authors · 2025-04-04
> > >
> > > Dear Reviewer QZbz –
> > >
> > > We sincerely appreciate your valuable comments and support. Our revision is updated to [anonymous.4open.science/r/eva-i](https://anonymous.4open.science/r/eva-i/README.md).
> > >
> > > ---
> > >
> > > **Summary.** We've updated Problem 1 as bilevel optimization (see [1-method](anonymous.4open.science/r/eva-i/revision-1-method.pdf)), where we carefully incorporated your feedback and cross-checked with several domain experts in algorithmic game theory. We've included new ablations along with all references you suggested (see [2-ablations](anonymous.4open.science/r/eva-i/revision-2-ablations.pdf)).
> > >
> > > ---
> > > 1. **Problem 1, The Game, and Regularization ([details](anonymous.4open.science/r/eva-i/revision-1-method.pdf))**
> > >
> > > We have revised Problem 1 to the bilevel setting below:
> > >
> > > $$
> > > \phi ^*  \in \underset{\phi}{\arg \max } \  R(\pi _\phi(\cdot) ; \pi _{\text {true}}(\cdot) ; \mathcal{D}, \theta ^* (\phi)) \\
> > > \textit{  s.t.} \quad \theta ^*  (\phi) \in \underset{\theta}{\arg \max} \ \mathbb{E} _{x \sim \pi _\phi(\cdot)}[\mathbb{E} _{y \sim \pi _{\theta}(\cdot | x)}[r(x, y)] - \beta \mathbb{D}[\pi _{\theta} \| \pi _{\text {base }}]].
> > > $$
> > >
> > > This naturally translates to a sequential game, where the inner is for the solver to optimize response alignment given the training prompt distribution, and the outter is for the creator to generate training prompts for the solver to perform well in the real world, knowing it will best respond.
> > >
> > > Here, $\pi_{\text {true }}$ is the true target prompt distribution, and $R(\cdot)$ is the "regularization" for creator. If $\pi_{\text {true }}$ is known, we can define $R(\cdot)$ to be some f-divergence measure. However, $\pi_{\text {true }}$ is often unknown a priori; this is then a standard decision under ignorance problem and the minimax regret rule gives a worst-case optimal solution. The optimization can be written as:
> > >
> > > $$
> > > \phi ^* \in \arg \max _\phi \ \text{Regret}(\pi _\phi, \pi _\theta) \\
> > > \textit{  s.t. } \quad \theta ^* (\phi) \in \arg \min _\theta \ \text{Regret}(\pi _\phi, \pi _\theta) .
> > > $$
> > >
> > > Note the inner loop optimization is equivalent. See the link above for details -- we believe the concerns on regularization should now be fully resolved. Please let us know if you'd like to see more explanations in the paper!
> > >
> > > ---
> > >
> > > 2. **Ablations ([details](anonymous.4open.science/r/eva-i/revision-2-ablations.pdf))**
> > >
> > > | Setting                   | $n _{\text{new}} = 4$ | $n _{\text{new}} = 8$ |
> > > |---------------------------|----------------------|----------------------|
> > > | RLOO (1x)                 | 52.6                 | 52.6                 |
> > > | RLOO-eva (1x)         | 57.3                 | **57.6**             |
> > > | RLOO-eva (2x)         | 60.5                 | **61.2**             |
> > > | RLOO-eva (3x)         | **62.4**             | **63.0**             |
> > >
> > >
> > > | Setting                   | ratio = 50% | ratio = 75% | ratio = 25% |
> > > |---------------------------|--------------------------------------|--------------------------------------|--------------------------------------|
> > > | RLOO (1x)                 | 52.6                                 | 52.6                                 | 52.6                                 |
> > > | RLOO-eva (1x)         | 57.3                                 | 57.0                                 | **57.5**                             |
> > > | RLOO-eva (2x)         | **60.5**                             | 59.9                                 | 59.2                                 |
> > > | RLOO-eva (3x)         | **62.4**                                 | 62.0                                 | 61.3                                 |
> > >
> > > We find increasing $n _{\text{new}}$ is helpful, and a balanced sampling is more robust. (Note online setting is an adaptation rather than the main contribution.)
> > >
> > > ---
> > >
> > > **Minor clarification.** We hope to clarify the ambiguity in our earlier response regarding the max-max and min-max part. Our initial intention was to clarify that there may not be formal definitions for competitive v.s. collaborative games; "competitive" adversarial training aims to get a more robust policy, which can be interpreted as "collaborative" too. (Side note: there are formal definitions on cooperative v.s. non-cooperative games, differ by whether players can communicate, commit and have binding agreement.) And if the optimization can be decoupled as $x ^*=\arg \max _x\{f(x, y ^*(x)) + R(x)\}, \textit{s.t. \ } y ^* (x)=\arg \max _y f(x, y)$, then replacing the inner max to be a min form may not alter the nature of the game. We appreciate your feedback and note our initial additive setting may be ambiguous, and have revised Problem 1 as discussed above.
> > >
> > >
> > > ---
> > >
> > > We believe the revision can sufficiently address previous concerns, and we sincerely hope you may re-consider the rating to `eva`. Please do let us know if you have any further comments, and we are more than happy to discuss with you. Thanks a lot!

---

### Official Review · Reviewer_eTrw · 2025-03-13

**Overall Recommendation:** 4

**Summary:**

The paper "Evolving Alignment via Asymmetric Self-Play" presents EVA, an innovative framework that addresses a critical limitation in current RLHF methods by replacing static prompt distributions with an adaptive, generative approach. By framing alignment as a game between a prompt creator and response solver, EVA enables continual self-improvement without requiring additional human data. The empirical results are impressive, boosting Gemma-2-9b-it's performance on Arena-Hard benchmarks significantly and allowing it to compete with much larger models. The method's seamless integration with both online and offline RLHF pipelines, coupled with its robust performance across different configurations, makes it a practical and valuable contribution to the field of language model alignment.


# after rebuttal

I will keep my score for accepting this paper.

**Claims And Evidence:**

Yes.

**Essential References Not Discussed:**

None

**Experimental Designs Or Analyses:**

Yes

**Methods And Evaluation Criteria:**

Yes.

**Other Comments Or Suggestions:**

N/A

**Other Strengths And Weaknesses:**

Strengths

1. Open-ended alignment is an important topic, given that the social trends and human opinions are evolved. This paper tackles this problem through the Evolving Alignment via Asymmetric Self-Play, i.e., EVA.

2. Though co-evolving of the creator and solver is promising, the training is unstable. This works tackle this issues by self-play and the regret-based methods. The framework is elegant and powerful.

3. The evaluation is sufficient and comprehensive.

Weakness

1. I am a bit concerning about the creator. If both creator and solver evolves, is it possible that they evolve to a bad local optimum?

2. Still about the game between the creator and the solver. Does the algorithm can find the equilibrium? or could you provide some analysis about this? As if building the problem as a game, the desired solution may be Nash equilibrium. so we can evaluation whether the formulation of the game is reasonable through checking whether the equilibrium is corresponding to the best solution.

**Questions For Authors:**

Please see above sections.

**Relation To Broader Scientific Literature:**

Relevant to the people who work on alignment, LLM and AI.

**Theoretical Claims:**

N/A

---

> ### Author Rebuttal · Authors · 2025-04-01
>
> We sincerely thank the reviewer's in-depth evaluation and new insights on the game. Below we provide a high-level summary with detailed rebuttal, and will add relevant discussions in the reivisions.
>
> ---
>
> **TL;DR**: Under reasonable assumptions, we can evolve to a *local* minimax optimum. (However, due to the nonconvex-nonconcave nature of the neural network optimization, global optimum is generally intractable to find.)
>
> ---
>
> > **Q:** *If both creator and solver evolves, is it possible that they evolve to a bad local optimum? Can the algorithm find the equilibrium, or could you provide some analysis about this?*
>
> **A:** To our knowledge, for the nonconvex-nonconcave minimax optimization problem, finding the global equilibrium is generally NP-hard. In sequential settings, there exist alternating gradient descent algorithms (Jin et al., 2020; Wang et al., 2019) that can achieve exact local convergence to *local* minimax. Thus yes, it is possible for them to be "bad" local minimax optimum that are far away from global minimax optimum.
>
> In simultaneous settings, recent works (Dennis et al., 2020; Parker-Holder et al., 2022; Beukman et al., 2024) have shown that when a Nash equilibrium is reached, the solver follows a minimax regret policy and the solution exhibits robustness properties.
>
> We believe the existing analysis helps justify the general game-theoretic formulation. In our empirical algorithm, we take the sequential setting, and use approximations for the regret maximization for the creator to avoid instability during training, as discussed in Section 3.2. We take a mixed sampling strategy to avoid the creator to drift too far away in each iteration (as discussed in the Appendix). Moving forward, we believe deriving a tractable algorithm with differentiable creators is a meaningful next step.
>
> ---
>
> **References**
> Jin, C., Netrapalli, P., & Jordan. M. (2020). What is Local Optimality in Nonconvex-Nonconcave Minimax Optimization.
> Wang, Y., Zhang G., & Ba. J. (2019). On Solving Minimax Optimization Locally: A Follow-the-Ridge Approach.
> Zhang, G. (2023). Deep Learning Dynamics: From Minimization to Games. Dissertation at University of Toronto.
> Dennis, M., Jaques, N., Vinitsky, E., Bayen, A., Russell, S., Critch, A., & Levine, S. (2020). Emergent complexity and zero-shot transfer via unsupervised environment design.
> Parker-Holder, J., Jiang, M., Dennis, M., Samvelyan, M., Foerster, J., Grefenstette, E., & Rocktäschel, T. (2022). Evolving curricula with regret-based environment design.
> Beukman, M., Coward, S., Matthews, M., Fellows, M., Jiang, M., Dennis, M., & Foerster, J. (2024). Refining minimax regret for unsupervised environment design.

---

### Official Review · Reviewer_8RPw · 2025-03-24

**Overall Recommendation:** 3

**Summary:**

The paper proposes Evolving Alignment via Asymmetric Self-Play (Eva), which treats post-training as an infinite game involving two roles: the creator, responsible for generating new prompts, and the solver, which optimizes responses. Eva implements prompt evolution via a regret-based reward objective combined with a prioritized generation buffer, which works for both online and offline RL training.
Eva shows strong performance on Arena-Hard, the win rate of gemma-2-9b-it increased from 51.6% to 60.1% (DPO) and 52.6% to 62.4% (RLOO).

**Claims And Evidence:**

Yes

**Essential References Not Discussed:**

This paper has adequately discussed relevant prior work.

**Experimental Designs Or Analyses:**

Yes.

Eva is evaluated across multiple RL algorithms:

- Online: RLOO, OAIF
- Offline: DPO, SPPO, etc. (§4)

Trained using UltraFeedback and tested on three benchmark evaluations, covering both online and offline RLHF scenarios. The choice of reward model (ARMORM-8B) is reasonable, supporting the validity of performance claims.

However, continuous training (§4.2.4) only reports monotonic gains, but does not analyze saturation points or changes in prompt quality after multiple iterations.

**Methods And Evaluation Criteria:**

Yes

**Other Comments Or Suggestions:**

See strengths and weaknesses.

**Other Strengths And Weaknesses:**

Strengths:
- The proposed Eva improves RL post-training performance without additional human prompts
- The empirical results are strong, gemma-2-9b-it’s win rate on Arena-Hard increased from 51.6% to 60.1% (DPO) and 52.6% to 62.4% (RLOO), surpassing Claude-3 Opus and approaching Gemini-1.5 Pro.
- Eva-generated curriculum prompts can outperform human prompts. Compared to a baseline using 6× more human prompts, Eva (1× prompts) performs better across multiple metrics.

Weaknesses:
- Table 4 shows that different approaches' performance are close.
- The initial prompt distribution (UltraFeedback) and the evolution process may be domain-specific (e.g., Figure 11 shows an imbalanced distribution in AlpacaEval), and cross-domain generalization was not fully tested.
- Table 16 shows that the evolved prompts mainly focus on technical tasks, therefore they may have limited coverage.

**Questions For Authors:**

See strengths and weaknesses.

**Relation To Broader Scientific Literature:**

None

**Theoretical Claims:**

Yes.

The open-ended RLHF objective (Problem 1) achieves continuous self-training by jointly optimizing prompts and response policies (§3). However, the authors acknowledge estimation bias but do not quantify its impact or provide error bounds.

---

### Decision · Program_Chairs · 2025-05-01

**Decision:**

Accept (poster)

**Comment:**

The paper proposes Eva, a scalable reinforcement post-training framework that replaces static human prompts with an adaptive, asymmetric self-play mechanism. Eva treats post-training as a game with a creator generating prompts and a solver optimizing responses. It also uses a regret-based reward objective and a prioritized generation buffer, working for both online and offline RL training. Evaluated on Arena-Hard and UltraFeedback datasets, Eva demonstrates significant performance gains for models like Gemma-2-9b-it and outperforms baselines using 6× more human prompts.

Reviewers acknowledge several strengths of the paper. As a practical, unified approach compatible with both online and offline RLHF pipelines, Eva improves RL post-training performance without extra human prompts. Empirical results via extensive experiments across RL algorithms and model scales show significant win-rate increases on the Arena-Hard benchmarks. The integration of regret-based optimization and prioritized buffers is also appreciated.

On the other hand, the paper has some drawbacks. Reviewer QZbz questions the scientific foundation, pointing out issues in game design and inconsistencies in problem formulation. Reviewer eTrw also worries about the creator evolving to a bad local optimum and the algorithm's ability to find an equilibrium. There are also concerns regarding the limited analysis of cross-domain generalization, heuristic hyperparameter choices, potential unfair comparisons with DPO baselines, and ambiguities in prompt buffer design.

The authors have submitted their responses to the reviews. Subsequently, all the reviewers have confirmed that they have read the authors' responses, and some of them have updated their overall assessments based on these responses.